



# 1 Insights on the water mean transit time in a high-elevation
# 2 tropical ecosystem

**G. M. Mosquera[1,2], C. Segura[3], K. B. Vaché[2], D. Windhorst[4], L. Breuer[4,5] and**
**Patricio Crespo[1]**
[1]{Departamento de Recursos Hídricos y Ciencias Ambientales & Facultad de Ciencias
Agropecuarias, Universidad de Cuenca, Av. 12 de Abril, Cuenca, Ecuador}
[2]{Department of Biological and Ecological Engineering, Oregon State University,
Corvallis, OR, USA}
[3]{Department of Forestry Engineering, Resources, and Management, Oregon State
University, Corvallis, Oregon, USA}
[4]{Institute for Landscape Ecology and Resources Management (ILR), Research Centre for
BioSystems, Land Use and Nutrition (IFZ), Justus Liebig University Gießen, Gießen,
Germany}
[5]{Centre for International Development and Environmental Research, Justus Liebig
University Gießen, Gießen, Germany}
Correspondence to: G. M. Mosquera. (giovanny.mosquerar@ucuenca.edu.ec)
**Abstract**
This study focuses on the investigation of the yet unknown mean transit time (MTT) of
stream waters and its spatial variability in tropical alpine ecosystems (wet Andean páramo).
The study site is the Zhurucay River Ecohydrological Observatory (7.53 km²) located in south
Ecuador. A lumped parameter model considering five transit time distribution (TTD)
functions was used to estimate MTTs. We used a unique data set of $\delta^{18}O$ and $\delta^2H$ isotopic
composition of rainfall and streamflow water samples collected for three years (May 2011-
May 2014) in a nested monitoring system of streams. Linear regression between MTT and





landscape (soil and vegetation cover, geology, and topography) and hydrometric (runoff
coefficient and specific discharge rates) variables was used to determine controls on MTT
variability, as well as mean electrical conductivity (MEC) as a possible proxy for MTT.
Results revealed that the exponential TTD function best describes the hydrology of the site,
indicating a relatively simple transition from rainfall water to the streams through the organic
horizon of the wet páramo soils. MTT of the streams is relatively short (0.15-0.73 yr, 53-264
days). Regression analysis revealed negative correlation between the catchment's average
slope and MTT ($R^2 = 0.78$, $p < 0.05$). MTT showed no significant correlation with
hydrometric variables whereas MEC increases with MTT ($R^2 = 0.89$ $p < 0.001$). Overall, we
conclude that: 1) MTT of streams confirms that the hydrology of the ecosystem is dominated
by shallow subsurface flow; 2) the interplay between the high storage capacity of the wet
páramo soils and the slope of the catchments provides the ecosystem with high regulation
capacity; and 3) MEC is an efficient predictor of MTT variability in this system of catchments
with relatively homogeneous geology.
Keywords: Ecohydrological processes, subsurface flow, mean transit time, lumped parameter
model, Andosol and Histosol, wet Andean páramo, tropical wetlands, South America

## 1   Introduction

Investigating ecohydrological processes by identifying fundamental catchment descriptors –
such as the MTT of waters – i.e., the average time elapsed since a water molecule enters a
catchment as recharge to when it exits it at some discharge point (Bethke and Johnson, 2002;
Etcheverry and Perrochet, 2000; Rodhe et al., 1996) – is fundamental in order to: 1) advance
global hydrological, ecological, and geochemical processes understanding and 2) improve the
management of water resources. This is particularly critical in high-elevation tropical
environments, such as the wet Andean páramo (further referred as "páramo"), in which,
hydrological knowledge remains limited, despite its importance as the major water provider
for more than 100 million people in the region (IUCN, 2002). Water originated from the
páramo sustains the socio-economic development in this region by fulfilling urban,
agricultural, industrial, and hydropower generation water needs (Célleri and Feyen, 2009).





Despite the importance of tropical biomes as natural sources and regulators of streamflow,
there are very few studies of MTT in tropical environments (e.g., Roa-García and Weiler,
2010; Timbe et al., 2014). The majority of MTT studies have been conducted in catchments
with strong climate seasonality, i.e., located in the northern and southern hemispheres (e.g.,
McGlynn and McDonnell, 2003; McGuire and McDonnell, 2006; McGuire et al., 2005), and
considerably less attention has been devoted to tropical environments. Most tracer-based
studies conducted in tropical latitudes focused on isotope hydrograph separation at storm
event scale (e.g., Goller et al., 2005; Muñoz-Villers and McDonnell, 2012), the isotopic
characterization of precipitation patterns (e.g., Vimeux et al., 2011; Windhorst et al., 2013),
and the identification of ecohydrological processes (e.g., Crespo et al., 2012; Goldsmith et al.,
2012; Mosquera et al., In review). However, studies focusing on MTTs in order to improve
the understanding of rainfall-runoff processes and their dependence on landscape biophysical
features in tropical regions are still lacking and urgently needed in order to improve water
resources management.
Prior investigations in our study site suggest that runoff originates from the shallow organic
horizon of the páramo soils located near the streams (Histosol soils or Andean wetlands), thus
favoring shallow subsurface flow; whereas deep groundwater contributions to discharge are
minimal (Mosquera et al., 2015, In review). The hydrological importance of shallow
subsurface flow to runoff generation has also been demonstrated in a variety of ecosystems
around the globe (e.g., Freeze, 1972; Hewlett, 1961; Penna et al., 2011), but yet, the MTT of
stream waters in these systems has not been explored in tropical regions. Therefore, our study
site provides a unique opportunity to gain understanding of the MTT of a shallow subsurface
flow dominated system in a tropical setting. In addition, the study of the MTT in natural
wetland systems has been limited to sites located in northern boreal catchments in Sweden
(Lyon et al., 2010), and peatlands, which are hydropedologically comparable to the Histosols
soils in our study site, in Scottish mountainous regions (e.g., Hrachowitz et al., 2009a;
Tetzlaff et al., 2014). Nevertheless, given that these catchments have significant contributions
from spring snowmelt and groundwater contributions to discharge, respectively, neither of
these allows for the isolation of the effect of wetlands in the subsurface transport of the water
within the catchments.



Another critical issue is the identification of controls on the MTT of stream waters. As
detailed observations of combined hydrometric and isotopic information are not feasible in
many regions due to limited funding and site accessibility, identifying controls of MTT
variability in nested and paired monitoring systems of streams is fundamental towards
regionalization of ecohydrological processes (Hrachowitz et al., 2009a) and prediction in
ungauged basins (Tetzlaff et al., 2010). Yet, investigation of controls on MTT variability is
still fairly scarce (Tetzlaff et al., 2013). Most studies have found that MTT scales with
topographic and/or hydropedological controls. For instance, topographical controls on MTT
variability were found in New Zealand catchments (Broxton et al., 2009) and a system of
streams in Oregon, USA (McGuire et al., 2005); whereas the proportion of wetlands and
responsive soils were reported as major MTT controls in Swedish catchments (Lyon et al.,
2010) and Scottish streams (Soulsby et al., 2006), respectively.
In this study, we seek to add to the current geographical scope of MTT studies by addressing
two questions which remain open in hydrological science and have received little attention in
high-elevation tropical ecosystems: "How old is stream water?" (McDonnell et al., 2010) and
"How does landscape structure influence catchment transit time across different geomorphic
provinces?" (Tetzlaff et al., 2009). Detailed hydrometric observations that highlighted
subsurface dominated rainfall-runoff response (Crespo et al., 2011; Mosquera et al., In
review) together with information of the landscape biophysical characteristics in our páramo
study site will allow for process-based understanding regarding: i) the spatial variability of
MTT and ii) the factors controlling such variability. Based on our current knowledge of the
hydrology of the ecosystem, i.e., apparent dominance of shallow subsurface flow to runoff
generation, we hypothesize relatively short MTTs of streams compared to systems dominated
by groundwater contributions to discharge. Also, based on the hydropedological and climatic
similarities between our páramo site and the peatland-podzols dominated ecosystems in the
Scottish highlands (e.g., Soulsby et al., 2006; Tetzlaff et al., 2014), we hypothesize the
proportion of wetlands to be a dominant control on the variability of the MTT in this high-
elevation tropical ecosystem.





**2   Materials and methods**
**2.1   Study site**
The Zhurucay River Ecohydrological Observatory is a basin located within a tropical alpine
biome, locally known as wet Andean páramo. It is situated in south Ecuador (3º04'S,
79º14'W) on the west slope of the Atlantic-Pacific continental divide and discharges into the
Jubones River (Pacific Ocean tributary). The basin has a drainage area of 7.53 km$^2$ and
extends within an elevation range of 3400 to 3900 m a.s.l. Climate is controlled by the Pacific
regime, although it is also influenced by the Amazonian regime to a lesser extent. Mean
annual precipitation at the observatory is 1345 mm at 3780 m a.s.l. Precipitation shows low
seasonallity with two relatively drier months (August and September) and primarilly falls as
drizzle (Padrón et al., 2015). Mean annual temperature is 6.0 °C at 3780 m a.s.l. and 9.2 °C at
3320 m a.s.l. (Córdova et al., 2015).
Geology primarily corresponds to volcanic rock deposits compacted by glacial activity during
the last ice age (Coltorti and Ollier, 2000). The Quimsacocha formation covers the northern
part of the basin and its lithology is composed by basaltic flows with plagioclases, feldspars,
and andesitic pyroclastics. The Turi formation covers the southern part of the catchment and
its lithology mainly corresponds to tuffaceous andesitic breccias, conglomerates, and
horizontally stratified sands. Both formations date from the late Miocene period (Pratt et al.,
1997). The geomorphology of the landscape bears the imprint of glaciated U-shaped valleys.
The average slope of the basin is 17%. The majority of the basin (72%) has mean gradients
between 0%-20%, although slopes up to 40% are also found (24%). There is an interesting
geomorphic feature in the northeastern side of the basin corresponding to a ponded wetland at
a flat hilltop. As indicated by geologists from INV metals mining company, this structure
most likely resulted from the eutrophication of a lagoon due to high accumulation of volcanic
material. This area is locally known as "Laguna Ciega" ("Blind Lagoon" in Spanish) and
drains towards the outlet of catchment M7 (see Figure 1). The analysis of the water stable
isotopic composition of soil water and streamflow in this area indicated that the hydrologic
processes of this site occur in the shallow ponded water that is directly connected to the





drainage network; while deeper water stored in the soil profile has little influence for
discharge generation most likely as a results of the eutrophic condition of the wetland
(Mosquera et al., In review).
Andosols, mainly found in the hillslopes, are the dominant soil type in the study site, covering
approximately 80% of the total basin area. Histosols (Andean wetlands), mainly found in flat
areas where rock geomorphology allows water accumulation cover the remaining portion of
the basin (Mosquera et al., 2015). These soils which have formed from the accumulation of
volcanic ash in flat valley bottoms and relatively low gradient slopes in combination with the
cold-humid climate have resulted in black, humic, and acid soils rich in organic matter
content with low bulk density and high water storage capacity (Quichimbo et al., 2012). The
organic fraction of the Histosol soils corresponds to an H horizon (median depth 76.5 cm);
while in the Andosol soils it corresponds to an Ah horizon (median depth 40cm). The mineral
fraction of both soils corresponds to a C horizon (median depth of 31 cm in the Histosols and
40 cm in the Andosols). A complete description of soil properties can be found in Mosquera
et al. (2015) and Quichimbo et al. (2012). Vegetation coverage is highly correlated with the
soil type. Cushion plants (such as *Plantago rigida, Xenophyllum humile, Azorella spp.*) grow
primarily in Histosols, while tussock grass (mainly *Calamagrostis* sp.) (Ramsay and Oxley,
1997; Sklenar and Jorgensen, 1999) grow in Andosols.
**2.2   Hydrometric information**
Discharge and precipitation were continuously monitored since October 2010. A nested
monitoring network was used to measure discharge. The network consisted of seven tributary
catchments (M1 to M7) draining to the outlet of the basin (M8). Catchments M1 to M6
comprise the main stream network draining towards the outlet of the Zhurucay basin (M8),
whereas catchment M7 is a small catchment originated from a ponded wetland at a flat hilltop
(Figure 1). V-notch weirs were constructed to measure discharge at the outlet of the
tributaries M1-M7 and a rectangular weir at the outlet of the basin M8. Each catchment was
instrumented with pressure transducers with a precision of ±5 mm. Water levels were
recorded at a 5-minute resolution, and transformed into discharge using the Kindsvater-Shen





relationship (U.S. Bureau of Reclamation, 2001). The discharge equations were calibrated by
applying the constant rate salt dissolution technique (Moore, 2004). Precipitation was
recorded using tipping buckets with a resolution of 0.2 mm at two stations located at 3780 and
3700 m a.s.l. (Figure 1).

## 2.3 Collection and analysis of water stable isotopic and electrical conductivity data

We used a three-year record (May 2011 – May 2014) of $^{18}$O and $^2$H isotopic compositions of
water samples collected in precipitation and streamflow. Data were collected at different
resolutions, from event-based to biweekly, given logistic constraints and opportunities.
Higher resolution data were aggregated to biweekly using precipitation amount weighted
means for record consistency. The same nested monitoring network used for measuring
discharge was implemented for measuring stable isotopes in streamflow (i.e., seven tributary
catchments M1 to M7 draining towards M8 at the outlet of the basin). Water samples in
precipitation were collected using two rain collectors located at 3780 and 3700 m a.s.l. Each
collector consisted of a circular funnel and a polypropylene bottled covered with aluminum
foil. Evaporation was prevented by placing a plastic sphere (4 cm diameter) in the funnel and
a layer of 0.5 cm mineral oil within the polypropylene bottle. Due to the sampling procedure
and the local climate, kinetic fractionation by evaporation can be neglected and hence both
stable isotopes yield the same results (Mosquera et al., In review). Therefore only the results
using the isotopic composition of $^{18}$O are reported. Rainwater samples are cumulative
representations of the isotopic signature between sampling dates while stream grab water
samples represent discrete points in time. The collected water samples were stored in 2 ml
amber glass bottles, covered with parafilm, and kept away from the sunlight to prevent
fractionation by evaporation as recommended by the International Atomic Energy Agency
(Mook, 2000). The isotopic composition of the water samples was measured using a cavity
ring-down spectrometer L1102-i (Picarro, USA) with a 0.5‰ precision for deuterium ($^2$H)
and 0.1‰ precision for oxygen-18 ($^{18}$O). Isotopic concentrations are presented in the δ
notation and expressed in per mill (‰) according to the Vienna Standard Mean Ocean Water
(V-SMOW) (Craig, 1961).





Electrical conductivity (EC) was measured directly instream simultaneously with the water
isotopic data starting in 2012, the second year of the monitoring period. EC was measured
using the digital conductivity sensor Tetracon 925 (WTW, Germany) with a precision of ±

4  0.5%.

**2.4  Mean transit time modeling and transit time distributions**
Mean transit time (MTT) was estimated using an inverse solution to the lumped parameter
model approach (Amin and Campana, 1996; Małoszewski and Zuber, 1982), which seeks for
the parameter set of the model that best describes the hydrologic system represented by a
predefined transit time distribution (TTD) function (Maloszewski and Zuber, 1996). The TTD
describes the transition of an input signal (e.g., precipitation, snow) of tracer (e.g., $\delta^{18}O$, $\delta^2H$)
to the signal at an outlet point (e.g., groundwater, streamflow) resulting from the subsurface
transport of water molecules within a catchment. Mathematically the TTD is described by a
convolution integral that transforms the input signal ($\delta_{in}$) into an output signal ($\delta_{out}$),
considering a time lag between both signals $(t - \tau)$ through a transfer function
(TTDs or $g(\tau)$) describing the subsurface transport of tracer as follows:

$$\delta_{out}(t) = \int_0^\infty g(\tau)\, \delta_{in}(t - \tau)\, d\tau \quad (1)$$

where $\tau$ is the integration variable representing the MTT of the tracer. A more robust
approximation weights the isotopic concentration of the input by considering recharge mass
variation ($w(\tau)$) so that the outflow composition reflects the mass flux leaving the catchment:

$$\delta_{out}(t) = \frac{\int_0^\infty g(\tau)\, w(t - \tau)\, \delta_{in}(t - \tau)\, d\tau}{\int_0^\infty g(\tau)\, w(t - \tau)\, d\tau} \quad (2)$$

where $w(t - \tau)$ can be described in terms of rainfall magnitude, intensity, or effective
precipitation (McGuire and McDonnell, 2006). Precipitation intensity was used to volume
weight the isotopic composition of precipitation in our study. Recharge was represented by
the rainfall isotopic composition weighted by precipitation rate and accounted for relatively





small recharge (i.e., lower precipitation inputs) during the less wet months (August and
September).
MTT ($\tau$ in Eqs. 1 and 2) was estimated by adjusting the response function or TTD to fit the
measured and simulated stream water isotopic composition. Five TTDs were considered to
investigate which better describes the subsurface transport of water molecules in the Zhurucay
basin. We used the exponential model (EM), exponential-piston flow model (EPM), the
dispersion model (DM) (Małoszewski and Zuber, 1982), the gamma model (GM) (Kirchner et
al., 2000), and the two parallel linear reservoir model (TPLR) (Weiler et al., 2003). Each
model is briefly described below and Table 1 summarizes their equations, fitting parameters,
and the range of initial parameters used in this study.
The EM represents a well-mixed system and assumes contributions from all flow paths. It
assumes a relatively simple transition of the tracer towards the stream network. The EPM is
an extension of the EM in which a delay in the shortest flow paths is assumed by the piston
flow portion of the system. In addition to the MTT, it has an additional fitting parameter ($\eta$),
which represents the ratio of the total volume to the volume represented by the exponential
distribution. The DM arises from the solution of the one-dimensional advection-dispersion
equation (Kreft and Zuber, 1978) and assumes that there is influence of hydrodynamic
dispersion in the system's flow paths. It also has two fitting parameters, the MTT and the
dispersion parameter ($Dp$), which relates to the tracer transport process. The GM is a more
flexible and general version of the exponential model in which the product of two parameters
provides an estimation of the MTT of the system. These parameters are the shape parameter
($\alpha$) and the scale parameter ($\beta$) (Kirchner et al., 2000). The TPLR represents two parallel
reservoirs each one represented by a single exponential distribution. It has three fitting
parameters, the MTT of the slow ($MTT_s$) and fast ($MTT_f$) reservoirs and a parameter
representing the fraction of each of them with respect to total flow ($\varphi$) (Weiler et al., 2003).
The MTT approach bases on the following assumptions: 1) the solute concentration is
conservative (i.e., the tracer does not react with other elements present in the system); 2) the
tracer concentration is measured in flux mode; 3) the tracer enters the system only once and





uniformly; 4) a representative tracer input can be identified; 5) transport of solute is one-
dimensional and represented by a single TTD; and 6) there is a uniform storage of water
within the catchment (i.e., steady state of the flow in the system) (Małoszewski and Zuber,
1982). The steady-state assumption is valid for humid environments during specific flow
characteristics (i.e., baseflow) (McGuire et al., 2002). In order to comply with the latter
assumption, streamflow water samples collected during extreme rainfall events were excluded
for the MTT simulations (McGuire and McDonnell, 2006; Muñoz-Villers et al., 2015).  To
obtain more stable results, we looped the available three years of isotopic data ten times
during calibration in order to extend the data series for 30 years as a warm-up period
following Hrachowitz et al. (2011) and Timbe et al. (2014).
**2.5   Model performance and uncertainty analysis**
The model performance was evaluated using the Kling–Gupta efficiency coefficient (KGE)
(Gupta et al., 2009). KGE ranges from $-\infty$ to 1, where unity indicates an ideal optimization.
KGE can be viewed from a multi-objective perspective because it accounts for correlation
(i.e., balancing dynamics, $r$), variability error ($\gamma$), and bias error ($\beta$) within a single objective
function. The efficiency is mathematically represented by the Euclidean distance ($ED$) in each
of the three dimensions ($r$, $\gamma$, and $\beta$) to an ideal point where all of them are maximized (i.e.,
where ideally the three factors are set to one). Efficiencies lower than 0.45 were considered
poor predictions (Timbe et al., 2014).
Depending on the TTD function used, 1 to 3 parameters were fitted during the simulations.
Each model was first run 10,000 times within a wide range of parameter values (Table 1).
Once a parameter value that yielded the best KGE was clearly identified, the model was run
again until obtaining 10,000 behavioral solutions (i.e., solutions corresponding to at least 95%
of the highest KGE) (Timbe et al., 2014) and their 5 and 95% limit bounds (i.e., 90%
confidence interval) were estimated using the Generalized Likelihood Uncertainty Estimation
methodology (GLUE) (Beven and Binley, 1992). In addition, the measure of identification
(MI) (Segura et al., 2012) was calculated as a metric of the model parameter identifiability.
The MI is defined as the ratio between the behavioral parameters range to the initial range and
indicates how well a parameter is identified. This metric is expressed as a percentage and by





definition, the smaller the value, the better the parameter identifiability. We considered a
parameter is well-identified if its MI is lower than 10%. The best model describing the
hydrologic conditions of the system was selected using the following criteria: 1) best
goodness of fit using the KGE criterion, 2) results that yielded the lower uncertainty
estimations, and 3) higher parameter identifiability using the MI criterion.
**2.6   Correlation analysis of MTT and catchment characteristics**
We used linear regression to investigate relations between landscape characteristics and
hydrological behavior with the MTT of the catchments. For this analysis, we included the
catchments which comprise the main drainage network (i.e., catchments M1 to M6) and the
catchment outlet (M8) given that they possess comparable hydropedological and
geomorphological characteristics. That is, catchments situated at the valley bottom have well-
defined interconnections between wetlands in the riparian areas and the surrounding Andosol
soils at the slopes. Catchment M7 on the other hand, is located at a flat hilltop at the outlet of
a wetland area which remains ponded throughout the year. The geomorphology of this
concave (lagoon shaped) structure and its ponded eutrophic condition has allowed for the
hydrologic processes to majorly occur in the shallowest ponded portion of the water directly
connected to the stream network (with little influence of the most likely immobile water
which remains stored in the deeper soil fraction) (Mosquera et al., In review). Therefore, its
hydrological response is not comparable to the other catchments where hydrologic processes
mainly occur in the soils and consequently was excluded from the regression analysis.
Statistical significance of the correlations was tested using the F-test at a 95% confidence
level (i.e., $p < 0.05$).
The landscape and hydrometric variables tested for correlation were obtained from previous
studies at the site (Mosquera et al., 2015) and from detailed soil, vegetation, and topographic
information provided by INV Metals. The landscape features considered were: soil type,
vegetation, geology, catchment size, slope, flow path length and gradient, and topographic
wetness index (TWI) (Beven and Kirkby, 1979) (Table 2). The hydrometric variables
considered were: annual runoff, annual precipitation, runoff coefficient, and streamflow rates





(Table 3). Weekly collected EC for three years (June 2012-June 2015) was averaged and also
tested for correlation with MTT.
**3    Results**
**3.1    Hydrologic and isotopic characterization in rainfall and streamflow**
Precipitation in the Zhurucay basin is evenly distributed throughout the year (Figure 2a),
except for two months with relatively lower precipitation inputs (i.e., August and September),
both accounting for less than 8% of total annual precipitation. The hydrograph at the outlet of
the basin (M8) also depicts a flashy response to precipitation inputs, even during these less
humid months (see zoom in Figure 2a). Similar behavior is observed at all catchments.
Spatially, annual precipitation (P) is evenly distributed across the basin with an average of
$1,275 \pm 9$ mm. Total annual runoff (Q) is spatially more heterogeneous, varying between 684
and 864 mm per year. Similarly, runoff coefficient (Q/P) shows relatively high spatial
variability between 0.55 and 0.68 (Table 3).
The $\delta^{18}O$ isotopic composition in rainfall is highly variable throughout the year (e.g., average
$-10.2 \pm 0.32$‰ at the upper station) (Figure 2b) and follows a seasonal pattern with
isotopically enriched values during highest precipitation rates (April-May), and isotopically
depleted values in the less humid period (August-September). The $\delta^{18}O$ isotopic composition
in streamflow collected during low flows on the other hand, is much more damped (average -
$10.0 \pm 0.06$‰, at M8) than the isotopic composition in precipitation (Table 4). The
relationship between the $\delta^2H$ and $\delta^{18}O$ isotopic composition in all catchments plots between
the global and local relationships in rainfall (Figure 3). However, there are differences in the
regions where they plot within the relationship. M7 plots in a larger region than all of the
other catchments (golden diamonds). M3, and M4 (bluish open circles, subgroup 1) plot lower
in the relationship than M1, M2, M5, M6, and M8 (reddish crosses, subgroup 2).
**3.2    Model selection and mean transit time evaluation**
In order to identify the TTD best suitable to describe the hydrologic system in the Zhurucay
basin, we tested and evaluated the performance of all TTDs at all catchments (only results for





M8, the basin outlet, are shown for brevity; however similar results were obtained for the rest
of the catchments). The MTTs for the best performing TTD for all catchments will be further
discussed in detail.
All TTDs reproduce the $\delta^{18}O$ isotopic composition at the outlet of the basin (M8) with
efficiencies varying between 0.50 and 0.76, i.e., above the threshold of model acceptance
(KGE > 0.45) (Table 5). The more flexible models, GM and TPLR yield the highest
performances with KGEs of 0.75 and 0.76, respectively. The EM and the EPM yield similar
efficiencies (KGE = 0.63), while the DM yields the lowest efficiency among all (KGE =
0.50). The models associated with the highest KGEs yield the highest uncertainty bounds
according to their threshold of behavioral solutions, likely explained by an inverse relation
between the number of fitting parameters in a given model and the span of the confidence
bands (Figure 4).
The models' parameter identification analysis indicates that even though the TPLR model
yields the highest KGE, the level of identification of its parameters is the poorest (Figure 5).
The identification metric (IM, i.e., the ratio of behavioral parameter range to the initial
parameter range, Table 5) yields high values for all parameters for the TPLR model (MIs
ranges between 29% and 85%). For the EPM, DM, and GM models one parameter is well
identified (MIs < 10%), while the others show higher MI values. The MI for the single
parameter that defines the EM is very strong (MI = 2%). This coupled analysis of model
efficiency and parameter identifiability indicates that although models with a higher number
of fitting parameters provide higher efficiencies, their parameters are more uncertain.
Taking into account both, the highest goodness of fit provided by a model and the
identifiability of its parameters, the EM is the model that best describes the temporal
variability of the $\delta^{18}O$ isotopic composition across the Zhurucay basin. Interpretation of
hydrologic processes is not feasible for all of the other models, as a result of the interplay
between different sets of poorly identified parameters.
The MTT probability density functions (PDFs, which indicate the distribution of MTTs in the
hydrologic system, Figure 6a) and cumulative density functions (CDFs, which express the
tracer ''mass recovery from an instantaneous, uniform tracer mass addition'', McGuire et al.,





2005, Figure 6b) for each of the models show that the EM and the EPM are clearly dominated
by short MTTs, indicating a prevalence of short and/or very rapid flow paths of water in the
subsurface. On the contrary, the other models depict a dominance of longer MTTs as
evidenced by long flattened tails at the end of their PDFs. In a system where the hydrology
appears to be dominated by shallow subsurface flow, such as the Zhurucay basin (Mosquera
et al., In review), long transit times are unlikely to occur. Therefore, the dominance of short
MTTs of the EM and EPM linked to short flow paths of water are more suitable to represent
the subsurface transport of water in these páramo soils. However, the $\eta$ proportional
parameter (the ratio of the total volume to the volume represented by exponential distribution)
of the EPM model is poorly identified. Thus, the EM is preferred to describe the hydrologic
conditions in Zhurucay basin. The EM was also found to describe the subsurface transport of
water of stream water in another volcanic soil dominated catchments in eastern Mexico
(Muñoz-Villers et al., 2015).
Hydrologically speaking, the EM represents a well-mixed reservoir with relatively simple
transition of the water (i.e., tracer) in the subsurface towards the stream network. Since the
isotopic composition of water stored in the Histosols (Andean wetlands) – which are
hydrologically connected to the drainage network – is homogeneous throughout the whole
profile of the organic and the shallow mineral horizon of this soil and there is no evidence of
significant groundwater contributions in the Zhurucay basin (Mosquera et al., In review), a
well-mixed reservoir is an appropriate assumption in the study site. In addition, as the
hydrologic system appears to be dominated by shallow subsurface flow within the porous
organic horizon of the páramo soils, a relatively simple transition of infiltrated precipitation
towards the catchment outlet is feasible. Therefore, this analysis, based on process
understanding and the characteristics of the physical system, in addition to the statistical
analysis of the hydrologic modeling, further support that the EM is the model that best
describes the transport of water across the Zhurucay basin.
Results of the EM for selected catchments with the longest (M3), intermediate (M6), and
lowest (M7) MTTs are shown in Figure 7 and statistics of the EM simulations at all
catchments are summarized in Table 6. The EM overcomes the modeling acceptance criterion
of KGE > 0.45 at all catchments with KGE values ranging between 0.48 and 0.84. The





longest MTTs are found in catchments M3 (0.73 years, 264 days) and M4 (0.67 years, 240
days) and the shortest at M7 (0.15 years, 53 days). The MTT for the other catchments vary
within this range. On average, within the 90% confidence level for the catchments forming
the main drainage network (M1-M6 and M8), MTT estimations show small variations (25
days at the lower confidence bound and 35 days at the upper confidence bound) with small
standard deviation (4 days for the upper bound and 6 days for the lower bound). For
catchment M7, variations are even smaller (9 days at the lower confidence bound and 11 days
at the upper confidence bound). In addition, the model performs best for catchments with high
variability in their isotopic composition during the monitoring period. For instance, catchment
M3 (Figure 7a) shows the smallest amplitude in isotopic variation for the observed and
simulated data (Table 6), coupled with the lowest KGE (0.48) and the highest MTT. On the
other hand, catchment M7 (Figure 7c) shows the highest amplitude in isotopic variation for
the observed and simulated data, coupled with the highest KGE (0.84) and the shortest MTT.
Similarly, catchment M6 (Figure 7b), which has a MTT shorter than the one in M3 and longer
than the one in M7, has an amplitude and KGE varying between the ones in M3 and M7. The
Monte Carlo simulations for the fitted parameter MTT (Figure 7) clearly depict how the MTT
which yield the highest KGE in each catchment decreases as the variation in their isotopic
composition increases as described above. Results from all the catchments are also described
by this trend.
The PDFs of the catchments (not shown for brevity) exhibit a dominance of relatively short
MTTs in the hydrology of the Zhurucay basin. The CDFs depict that the tracer is completely
recovered in all catchments at around 80 biweeks, except for M7, where the tracers is even
more rapidly recovered (~ 19 biweeks). As we used a stable isotopic record of 78 weeks (3
years), these results indicate that a three years record of tracer data is enough to estimate the
MTT of waters using the LPM approach in the páramo basin of the Zhurucay observatory.

### 26    3.3    Correlations of Mean Transit Time with landscape and hydrometric
### 27    variables

Correlation analysis showed no statistically significant correlations (p-values > 0.05) with
landscape features and hydrometric variables of the nested monitoring system when all




catchments were included. This lack of correlation is likely related to the previously reported
distinct responsiveness of catchment M7 to precipitation inputs due to its different
geomorphology (i.e., ponded eutriphozed wetland disconnected from the slopes) and
catchments M3 and M4 (Figure 1) driven by a spring water contribution during low flow
generation. Although in general, groundwater contributions to discharge seem to be minimal
and geology has not been found to directly control the hydrology in this páramo ecosystem
(Mosquera et al., 2015), the existence of this shallow spring sourced at the interface between
the soil mineral horizon and the shallow bedrock upstream the outlet of M3 and M4 –
favoring the generation of higher low flows (Mosquera et al., In review) and increasing MTTs
in these catchments – indicates that geology (fractures in the shallow bedrock) influence the
hydrology of these small headwater catchments; thus, masking relationships between
landscape features and MTT of the whole system. Therefore, we tested the MTT correlations
without including these small catchments (M3 and M4) and M7 (subgroup 2). The reanalysis
with the modified data set revealed significant relations of MTT with topographical indexes
(Figure 8). The relations between MTT and average slope (Figure 8a, $R^2 = 0.78$ and p =
0.047) and percent area having slopes in the range 20%-40% are negative (Figure 8b, $R^2 =$
0.90, p = 0.015). Conversely the relation between the percent area having slopes 0%-20% and
MTT is positive ($R^2 = 0.85$, p = 0.026).
The regression analysis including all catchments also showed that mean electrical
conductivity (MEC) of the waters explains 89% (p = < 0.001) of the catchments' MTT
variability (Figure 9). Streams with higher MEC have longer MTTs. No significant
correlations (p-values > 0.05) between MTT and vegetation, soil types, geology, flow path
length, and topographic wetness index were found (Table 7).

## 4  Discussion

### 4.1  General hydrometric and isotopic characterization

The rainfall-runoff process evidences a rapid response of discharge to precipitation inputs in
the Zhurucay basin. This rapid response occurs even during the less humid periods (August-
September) in which relatively small rainfall events result in peak flow generation (Figure



2a). This high responsiveness results from the combined effect of the relatively uniform
distribution of precipitation year-round – common in tropical regions – and the unique
properties of the Histosol soils or Andean wetlands located near the streams. The high storage
capacity of wetlands was also highlighted by (Roa-García and Weiler, 2010) after the
comparison of three paired catchments in the growing coffee region of Colombia at lower
elevations (2000-2200 m a.s.l.). Similarly, Histosol soils in our study site are rich in organic
matter content (mean 86% by volume), allowing for high water storage capacity. In addition,
due to their relatively low saturated hydraulic conductivity (0.72-1.55 cm h$^{-1}$), these soils
remain near saturation throughout the year. These factors, in combination with the local
climate, allow páramo soils to regulate and maintain a sustained discharge throughout the
year. Moreover, as these processes occur in the shallow organic horizon of the soils, the
hydrology of the Zhurucay basin páramo ecosystem is dominated by shallow subsurface water
flow. This is supported by the similar isotopic composition between streams and soil waters in
the organic layer of the Histosols in the Zhurucay basin (Mosquera et al., In review).
The $\delta^{18}$O isotopic composition of stream waters is damped and lagged with respect to that of
precipitation. Nevertheless, streamflow samples in the Zhurucay basin still reflect the
variability of the $\delta^{18}$O composition of rainfall (Figure 2b), as expected in a system dominated
by shallow subsurface flow. The relationship between the $\delta^2$H and $\delta^{18}$O isotopic composition
in rainfall (LMWL) and streamflow indicates no isotopic fractionation by evaporation in local
precipitation and streamflow (see Mosquera et al., In review for details). Differences in the
region where catchments plot within this relationship indicate that they are differently
influenced by precipitation. M7, located at the outlet of a wetland that remains constantly
ponded, shows a faster response to rainfall, most likely as a result of the rapid mixing of
rainfall water with the shallow water moving in the organic horizon of the soils and the
ponded water above it. All of the other catchments show considerably less influence of
rainfall, although M3 and M4 (subgroup 1) show depleted values than M1, M2, M5, M6, and
M8 (subgroup 2). The latter most likely reflecting a small contribution from a small shallow
spring source to subgroup 1 (Mosquera et al., 2015).





## 4.2 What is the MTT of stream waters?

The high performance (KGE > 0.48) of the exponential model (EM) and its strong parameter
identification (Table 6) indicate that this model best mimics the subsurface transport of water
in all catchments within the Zhurucay basin (Figure 7). Nevertheless, the model captures
some particularities in the functioning of each catchment. For instance, results indicate
relatively long MTTs in two of the headwater catchments, M3 and M4 (0.73 and 0.67 years,
respectively). This results from a shallow spring water contribution to these catchments
during low flow generation (Mosquera et al., 2015). The model seems to capture the effect of
the shallow spring contribution by yielding the longest MTTs estimations in these catchments,
and an intrinsic influence of geology on MTT variability. In addition, the performance of the
model in these two catchments is the lowest within the basin. The latter most likely because
of less efficient mixing of water due to the influence of the spring water source; suggesting
that this effect is also captured by the model which assumes a well-mixed reservoir. In this
sense, it seemed logic to consider that another model representing an additional slow reservoir
(e.g., TPLR or GM) could have better represented the subsurface movement of water in these
catchments. Nevertheless, our results suggest that the contribution from this additional water
source is small and an additional reservoir is not well distinguished by these TTDs as their
parameters are not well identified. Recently, Muñoz-Villers et al. (2015) also identified the
EM as the model that best mimics subsurface flow in 7 of 12 nested catchments underlain by
volcanic soils (Andosols) in a TMCF located in Veracruz, Mexico. These authors estimated
even longer MTTs (1.2-2.2 years) due to deeper groundwater contributions to discharge.

On the other end, M7, dominated by contribution from the shallowest part of the organic
horizon of the soils and the ponded fraction of water accumulated in a ponded wetland –
which is directly connected to the stream channel – presents the shortest MTT of all
catchments (0.15 years, 53 days), linked to the highest model performance. Our results
support the hypothesis that this catchment presents a shorter MTT, indicating that the ponded
condition of the wetland allows for a rapid and efficient mixture of precipitated water with
ponded water and water stored in and released from the shallow organic horizon of the soil.
The latter resulting in a rapid delivery of event (new) water to the stream; whereas water
stored deeper in the soil seems to remains mostly immobile with minimal influence in the





hydrology of the catchment. The well-mixing and simpler delivery of water to the stream is
also captured by the high model performance.
The MTTs estimated for the rest of the catchments lie in between these two extremes and
their values and efficiencies vary depending on the amplitude of the isotopic tracer variation,
with longer MTTs in catchments were the amplitude of the signal is more damped –
evidencing lower influence of precipitation and less efficient mixing with the soil storage –
and vice versa. The MTT in these catchments vary relatively little in comparison to the rest of
the catchments (0.43 to 0.53 years, 156 to 191 days). Overall, the MTTs are relatively short,
further supporting previous evidence that shallow subsurface flow dominates the hydrology of
the ecosystem.
In other tropical latitudes, MTTs higher than 300 days were found in three paired Colombian
catchments applying the TPLR model (Roa-García and Weiler, 2010). These basins show
higher MTTs than the catchments in the Zhurucay basin most likely as a result of the higher
development of the volcanic ash soils (> 10 m), which allow the water to be stored for longer
periods in the subsurface. MTTs of stream waters longer than two years were also found in a
tropical montane cloud forest (TMCF) in southern Ecuador (Timbe et al., 2014), evidencing
that differently from our findings, this lower elevation ecosystem is dominated by deep
groundwater contributions. Preliminary MTT estimations of stream water in another TMCF
biome located in central Mexico (Muñoz-Villers and McDonnell, 2012) yielded a MTT of
three years. Although the ecosystem is dominated by soils formed by volcanic ash
accumulation, as the páramo soils are, a combination of deeper hillslope soils (1.5-3 m depth)
with highly fractured and permeable geology allows for the formation of longer flow paths of
water and longer MTTs. Therefore, the relatively young and little weathered geology in the
Zhurucay basin allows for a dominance of shallow subsurface flows. The results of these
studies suggest that the particular shallow development of the rich organic soils with low
saturated hydraulic conductivities, in combination with an homogeneous and low permeable
geology provide the páramo basin of the Zhurucay River with a high water retention capacity,
and relatively long transit times and flow paths considering the little development of the
organic horizon of the soils. Hrachowitz et al. (2009b) reported MTT of stream water (135-
202 days) around the ones found in the Zhurucay basin catchments in a montane catchment in





Scotland dominated by peatland soils and relatively little weathering geology. Nevertheless,
the models which provided the best fit were the GM and the TPLR, as opposed to the EM in
our study site. As in the Zhurucay basin, these authors attributed this short transit time to the
dominance of ecohydrological processes occurring in the upper horizon of the peat soils.
Therefore, we can conclude that in these two ecosystems, located at different latitudes but
with similar hydropedological conditions, the hydrology is dominated by shallow subsurface
flows. Nevertheless, the soils development of the shallow peaty soils in Scotland is lower (40
cm) in comparison to the soil development of the Histosols (80 cm) in the Zhurucay basin.
These factors, in combination with differences underlying geologies suggest that their overall
hydrologic functioning might differ as evidenced by different TTDs describing the subsurface
transport of solute.
**4.3  Controls on MTT variability**
We found significant correlations ($R^2 \geq 0.78$, $p < 0.05$) between catchment slope dependent
indexes and MTT using a subset of the main stream catchments (subgroup 2) (Table 7, Figure
8). Results of the correlation analysis indicate that 1) the higher the average slope of the
catchments, the shorter the MTT; 2) the higher the percent of area corresponding to slopes
between 0% and 20%, the longer the MTT; and 3) the higher the percent of area
corresponding to slopes between 20% and 40%, the shorter the MTT. These results indicate a
clear control of the catchments' slopes in the MTT of stream waters in the Zhurucay basin.
Locally, the same topographical features were found to control low flow generation.
Mosquera et al. (2015) attributed the latter to expected contributions from the water originated
in the slopes (Andosol soils) during low flow generation as a result of the gravitational
potential of the water that drains downslope from these soils. These authors also found that
wetlands (Histosols soils located near the streams) control the generation of moderate and
high flows. Although we did not find significant correlations with other landscape features,
vegetation shown expected trends in relation to MTT. That is, catchments with higher
proportion of cushion plants (wetlands) ($R^2 = 0.29$, $p = 0.35$) have longer MTTs and an
inverse relation with tussock grass vegetation ($R^2 = 0.31$, $p = 0.33$). In another tropical system
of catchments in Colombia, a catchment with higher areal proportion of wetlands was found



to prolong the MTT of stream waters, but appeared to reduce water yield (Roa-García et al.,
2011). Although these authors did not report the slope of the catchments, we can infer that the
catchment with the highest proportion of wetlands – as they form in flat areas – is also the
catchment with the lowest gradients. Therefore, their observations might result from the
combination of the deeper soil development (> 10 m) with high water retention capacity and
low saturated hydraulic conductivity, perhaps in combination with low slope gradients. This
would support the result of our study, where the catchments with the lower slopes and higher
proportion of wetlands present the longer MTTs.
In other latitudes, in 20 Scottish catchments with different geomorphologies and climate,
MTT variability was controlled by the areal proportion of peat soils and no influence of
catchments' slopes was found (Hrachowitz et al., 2009a). As such, and given the similarities
between these soils and our Histosol soils (Andean wetlands), we hypothesized the MTT
variability of streams to be controlled by the areal extent of wetlands. Even though we found
that MTT variability is rather majorly controlled by topography in our tropical alpine site, a
small trend of wetlands' cover to increase MTT was also identified. Although the later
relation is not statistically significant, the latter most likely results from the influence of
topography on Histosol soils (wetlands) formation, where the formation of this soil mainly
occurs in catchments with lower slopes where water accumulation is favored. This finding
indirectly suggests that wetlands influence MTT spatial variability to a lesser extent.
Therefore, it appears that although relatively similar processes control the ecohydrology of
both ecosystems, controls on MTT variability cannot be extended from one ecosystem to the
other. MTT variability was also found to be controlled by the proportion of wetlands in cold
snow dominate boreal catchments in Sweden for the MTT of spring snowmelt water (Lyon et
al., 2010). These authors attributed this effect to the formation of shallow ice acting as
impermeable barriers above the wetlands, and thus changing the flow paths of water.
Nevertheless, because of the different climate and geological features between their
catchments and ours, we did not find wetlands as major controls on MTT variability.
Other slope topographic indexes – e.g., flow path length (L), flow path gradient (G), and the
ratio between both (L/G) (e.g., McGuire et al., 2005; Tetzlaff et al., 2009) – have been
identified to control MTT variability in catchments in other latitudes. Although these




landscape features did not significantly explained MTT variability in the Zhurucay basin, the
L/G ratio was reported as the major control of MTT variability ($R^2 = 0.91$) in steep temperate
catchments in the central western Cascades of Oregon (McGuire et al., 2005), suggesting that
this relation "reflects the hydraulic driving force of catchment-scale transport (i.e., Darcy's
law)". Similarly to our study site, they also found average slope of these catchments to be one
of the most important individual controls on MTT, explaining 78% of the MTT variability.
Recently, topography was also identified as a major control on the MTT of 12 TMCF
catchments in eastern Mexico (Muñoz-Villers et al., 2015). Results from our these two studies
reflect that the integrated effect of catchment slope on MTT variation can be identified in
distinct geological and hydropedological provinces. The latter also suggests that rather than
using a predictor which indicates more local effects of hydraulic force driving in the stream
channel (e.g., L/G), catchment slope might be a better measure to compare catchment
functioning as it integrates the hydrologic connectivity of hillslope, riparian, and stream areas.
The catchment slope topographic controls on MTT in the Zhurucay basin indicate that water
resides for a longer time in the hydrologic system of catchments having lower slope gradients.
These results also indicate that in catchments having higher areal proportions of low gradients
and lower areal proportions of steeper gradients coupled with higher wetlands coverage, water
resides longer in the shallow reservoir of the soils. Therefore, it is apparent that water stored
in the wetlands is released to the streams depending on the catchments' topography. In
addition, the control of the proportion of steeper gradients in MTT variability also suggests
that the gravitational potential of water draining downslope in the Andosol soils also
indirectly influences the MTT of the streams. Overall, these results indicate that the high
storage capacity of the wetland soils (Histosols) located near the streams is not the only factor
providing páramo ecosystems with a high regulation capacity. Rather, it seems that the
interplay between the high storage capacity of the wetlands and the topography of the terrain
is what drives the extremely high water regulation capacity of this ecosystem. These
interpretations do not only make physical sense, but also add to our current process-based
understanding of páramo hydrology. In this shallow subsurface flow dominated system with a
high soil water retention capacity, it is clear that catchment topography is the factor driving
water movement. Without the interplay storage-slope, water would remain stored in the soils,



and perhaps the delivery of water towards the streams would be dominated by saturated
overland flow (SOF), affecting the regulation capacity of the ecosystem. Nevertheless, SOF
rarely occurs in the Zhurucay basin (Mosquera et al., 2015). Therefore, it is our interpretation
that the hydrology of this ecosystem is mainly dominated by two factors: 1) the high storage
capacity in the shallow organic horizon of the porous páramo soils and 2) the catchment
slope. Factor 1 driving the high water retention capacity and factor 2 controlling the high
regulation capacity of the ecosystem, and thus, maintaining a sustained delivery of water to
the streams along the year.
Mean electrical conductivity (MEC) was also found to be significantly correlated with the
MTT of the streams using all catchments of the nested system in the basin (Figure 9). The
regression analysis, showed strong correlation, with MEC increasing as the MTT of water
increases. As EC is an intrinsic property of water, due to the time it spends in contact with the
surrounding pore space, rather than a control on MTT variability, this result indicates that this
property might be used as a proxy to estimate MTT spatial variability. The well-defined
connection between MTT and MEC most likely resulting from the relatively homogenous
geology of the Zhurucay basin. To our knowledge, there are no studies that have identified
similar (or different) relations between MEC and MTT in other biomes.
Given that estimating MTT using isotope tracers and the LPM approach is financially
expensive due to the logistical set up of a monitoring network and the processes of data
collection and analysis, finding proxies (i.e., predictors) which allow inferring the MTT of
stream waters at lower operational costs is critical to improve water resources management. In
this sense, the strong relation between MEC and MTT indicates that MEC could be used as a
relatively inexpensive and directly measurable proxy for MTT in this wet Andean páramo
catchment. Therefore, although this result cannot be expanded beyond páramo areas, perhaps
not even beyond the study site, it seems that it is worth evaluating whether or not MEC can
infer MTT in other hydrologic systems. Nevertheless, one should be careful that EC
measurements can be relatively variable over time. As a result, a single measurement of EC is
most likely not enough to provide robust MTT estimates. Therefore, the longer the record of
EC measurements, the smallest the variability of MEC and the highest the robustness of MTT
estimates.



**5  Conclusions**
The MTT evaluation using a LPM indicated that the EM best describes the subsurface
transport of water in the basin. This result indicates efficient mixing in the high organic and
porous wet Andean páramo soils and a simple subsurface transition of rainfall water towards
the streams. MTT estimations showed relatively short MTTs of stream waters linked to
relatively short subsurface flow paths. Therefore, we confirm that the hydrologic system of
the tropical alpine biome of the Zhurucay basin is dominated by shallow subsurface flow.
MTT estimations showed that catchment M7, located at a flat hilltop at the outlet of a wetland
which remains ponded year-round and disconnected from the slopes – most likely as a result
of the eutrophication of a lagoon – showed a particularly low MTT (0.15 yr – 53 days) in
relation to the MTT in all of the other catchments (0.40-73 yr, 156-250 days) in which the
morphology corresponds to U-shaped valleys, with the wetlands located at the valley bottoms
near the streams and connected to the slopes. Two headwater catchments, M3 and M4,
showed the longest MTT, related to a small contribution from a spring shallowly sourced.
These results indicate that in this páramo ecosystem, the geomorphology of the wetlands and
geology to a lesser extent, influence the responsiveness of the streams to precipitation inputs.
Correlation analysis between landscape variables and MTT indicates that MTT variability is
majorly explained by the slope of the catchments, and a related influence of vegetation to a
lesser extent. Catchments with the steepest average slopes and lower proportion of wetlands
have the shortest MTTs. The lack of significant correlations between the MTT of streams and
hydrological response variables (runoff coefficient and specific discharge rates) indicate that
neither water yield, nor streamflow rates control the time water resides in subsurface of the
páramo soils. These results indicate that the interplay between the high storage capacity of the
páramo soils and the slope of the catchments define the ecosystem's high regulation capacity.
Mean electrical conductivity (MEC) of stream waters – with the oldest waters presenting the
highest MECs – seems to be a promising proxy of MTT in system of catchments under
homogeneous geological conditions. Finally, we want to highlight the usefulness of a nested
monitoring system for acquiring better process-based hydrologic functioning understanding.
For instance, if M3, M4, and/or M7 catchments would not have been monitored, the influence
of geology and/or geomorphology on catchment hydrological response could not have been



identified and important information about the whole ecosystem functioning would remain
unknown.
**Acknowledgements**
The research was funded by the Central Research Office at the University of Cuenca (DIUC)
via the project "Desarrollo de indicadores eco-hidrológicos funcionales para evaluar la
influencia de las laderas y humedales en una cuenca de páramo húmedo"; and the Ecuadorian
National Secretariat of Higher Education, Science, Technology and Innovation (SENESCYT).
The authors express gratitude to INV Metals S.A. (Loma Larga Project) for the logistic
support. Special thanks to Alicia Correa for her fieldwork assistance and Irene Cardenas for
the laboratory analysis.





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



Table 1. Models considered to describe mean transit time (MTT) of stream waters in the study
area and their transit time distribution (TTD) functions, parameters, and range of initial
parameters.

| Model | Transit time distribution ($g(\tau)$) | Parameter(s) range |
|---|---|---|
| Exponential model (EM) | $\dfrac{1}{\tau}\exp\left(\dfrac{-t}{\tau}\right)$ | $\tau\ [0-200]$ |
| Exponential-piston model (EPM) | $\dfrac{\eta}{\tau}\exp\left(-\dfrac{t\cdot\eta}{\tau}+\eta-1\right)$ for $t\geq\tau\left(1-\eta^{-1}\right)$ | $\tau\ [0-200]$<br>$\eta\ [0.5-4]$ |
| Dispersion model (DM) | $\left(\dfrac{4\pi D_p t}{\tau}\right)^{-1/2}t^{-1}\exp\left[-\left(1-\dfrac{t}{\tau}\right)^2\left(\dfrac{\tau}{4D_p t}\right)\right]$ | $\tau\ [0-200]$<br>$D_p\ [0.5-4]$ |
| Gamma model (GM) | $\dfrac{\tau^{\alpha-1}}{\beta^\alpha\Gamma(\alpha)}\exp^{-\tau/\beta}$ | $\tau\ [0-200]$<br>$\alpha\ [0.5-4]$<br>$\beta=\ \tau/\alpha$ |
| Two parallel linear reservoir (TPLR) | $\dfrac{\varphi}{\tau_f}\exp\left(\dfrac{-t}{\tau_f}\right)+\dfrac{1-\varphi}{\tau_f}\exp\left(\dfrac{-t}{\tau_s}\right)$ | $\tau_s\ [0-200]$<br>$\tau_f\ [0-20]$<br>$\varphi\ [0-1]$ |

$\tau$ = tracer's mean transit time (MTT) in biweeks; $\eta$ = parameter that indicates the percentage
of contribution of each flow type; $Dp$ = dispersion parameter; $\tau_f$ and $\tau_s$ = transit time of fast
and slow flows in biweeks; $\varphi$ = flow partition parameter between fast and slow flow
reservoirs.





Table 2. Main landscape characteristics of the monitored catchments.

| Catchment | Area (km²) | Altitude (m a.s.l) | Distribution of soil types[a] (%) | | | Vegetation cover[b] (%) | | | | Topography[c] (%) | | | | Geology[e] (%) | | | EC[f] (µS/cm) |
|---|---|---|---|---|---|---|---|---|---|---|---|---|---|---|---|---|---|
| | | | AN | HS | LP | TG | CP | QF | PF | AS | L[d] | G | TWI | Qm | Tu | Qd | |
| M1 | 0.20 | 3777 – 3900 | 85 | 13 | 2 | 85 | 15 | 0 | 0 | 14 | 0.9 | 1.9 | 9.6 | 100 | 0 | 0 | 35.7 |
| M2 | 0.38 | 3770 – 3900 | 83 | 15 | 2 | 87 | 13 | 0 | 0 | 24 | 0.8 | 1.9 | 12.8 | 66 | 1 | 33 | 32.0 |
| M3 | 0.38 | 3723 – 3850 | 80 | 16 | 3 | 78 | 18 | 4 | 0 | 19 | 1.0 | 2.0 | 9.4 | 59 | 41 | 0 | 62.4 |
| M4 | 0.65 | 3715 – 3850 | 76 | 20 | 4 | 79 | 18 | 3 | 0 | 18 | 1.3 | 2.0 | 12.5 | 50 | 48 | 1 | 47.9 |
| M5 | 1.40 | 3680 – 3900 | 78 | 20 | 2 | 78 | 17 | 0 | 4 | 20 | 2.5 | 1.9 | 11.8 | 70 | 1 | 30 | 37.0 |
| M6 | 3.28 | 3676 – 3900 | 74 | 22 | 4 | 73 | 24 | 1 | 2 | 18 | 3.3 | 1.8 | 8.2 | 50 | 30 | 20 | 35.6 |
| M7 | 1.22 | 3771 – 3830 | 37 | 59 | 4 | 35 | 65 | 0 | 0 | 12 | 0.4 | 1.7 | 10 | 87 | 0 | 13 | 15.3 |
| M8 | 7.53 | 3505 – 3900 | 72 | 24 | 5 | 71 | 24 | 2 | 2 | 17 | 4.6 | 1.9 | 16.7 | 56 | 31 | 13 | 33.5 |

[a] AN = Andosol; HS = Histosol; LP = Leptosol

[b] TG = tussock grasses; CP = cushion plants; QF = *Polylepys forest*; PF = pine forest.

[c] AS = average slope, L = flow path length ; G = flow path gradient; TWI = topographic wetness index (Beven and Kirkby, 1979).

[d] L units in km.

[e] Qm = Quimsacocha formation; Tu = Turi formation; Qd = Quaternary deposits.

[f] EC = mean electrical conductivity. Data collected weekly for a three years period (June 2012-June2015)



Table 3. Main hydrometric variables of the catchments.

| Catchment | Precipitation (mm yr$^{-1}$) | Total runoff (mm yr$^{-1}$) | Runoff Coefficient[b] | Average specific discharge (1 s$^{-1}$ km$^{-2}$) | Flow rates as frequency of non-exceedance (1 s$^{-1}$ km$^{-2}$) | | | | | | | |
| | | | | | $Q_{min}$ | $Q_{10}$ | $Q_{30}$ | $Q_{50}$ | $Q_{70}$ | $Q_{90}$ | $Q_{max}$ |
| M1 | 1300 | 729 | 0.56 | 23.1 | 0.7 | 2.7 | 6.6 | 14.3 | 26.4 | 50.1 | 1039.0 |
| M2 | 1300 | 720 | 0.55 | 22.8 | 1.2 | 4.8 | 7.9 | 14.9 | 26.7 | 49.0 | 762.9 |
| M3 | 1293 | 841 | 0.65 | 26.7 | 2.3 | 7.3 | 10.8 | 17.7 | 28.1 | 52.4 | 894.2 |
| M4 | 1294 | 809 | 0.62 | 25.6 | 4.2 | 6.2 | 9.8 | 16.6 | 27.3 | 52.1 | 741.2 |
| M5 | 1267 | 766 | 0.6 | 24.3 | 1.5 | 4.1 | 8.3 | 15.3 | 26.9 | 50.8 | 905.7 |
| M6 | 1254 | 786 | 0.63 | 24.9 | 1.2 | 3.7 | 8.2 | 15.9 | 27.5 | 53.2 | 930.4 |
| M7 | 1231 | 684 | 0.56 | 21.7 | 0.3 | 1.8 | 5.2 | 11.0 | 23.3 | 53.9 | 732.0 |
| M8 | 1277 | 864 | 0.68 | 27.4 | 1.9 | 4.0 | 8.7 | 15.2 | 29.2 | 60.8 | 777.9 |

[a] Total runoff as a proportion of precipitation.





Table 4. Statistics of the δ18O isotopic composition in precipitation and streamflow used as
input data for the MTT modeling.

| Sampling Station | Altitude (m a.s.l.) | δ18O (‰) | | | | |
|---|---|---|---|---|---|---|
| | | n[a] | Average | SE[b] | Max | Min |
| M1 | 3840 | 123 | −10.6 | 0.06 | −9.0 | −12.6 |
| M2 | 3840 | 124 | −10.4 | 0.07 | −8.8 | −12.6 |
| M3 | 3800 | 121 | −10.7 | 0.05 | −8.8 | −12.1 |
| M4 | 3800 | 122 | −10.6 | 0.05 | −8.7 | −11.9 |
| M5 | 3800 | 118 | −10.5 | 0.06 | −9.1 | −12.8 |
| M6 | 3780 | 121 | −10.3 | 0.06 | −8.9 | −12.2 |
| M7 | 3820 | 121 | −8.9 | 0.15 | −6.2 | −13.9 |
| M8 | 3700 | 118 | −10.0 | 0.06 | −8.3 | −11.6 |
| Upper Precip. | 3779 | 137 | −10.2 | 0.32 | −1.2 | −25.0 |
| Middle Precip. | 3700 | 134 | −10.1 | 0.32 | −2.7 | −20.0 |

[a] n: number of samples collected.
[b] SE: Standard error.

Table 5. Statistical parameters of observed and modeled δ¹⁸O for the stream at the outlet of the basin (M8)

| Model | Observed δ¹⁸O Mean (‰) | σ[a] (‰) | Simulated δ¹⁸O Mean (‰) | σ[a] (‰) | KGE[a] (-) | Model Parameters[b] | | MI[c] (%) |
|---|---|---|---|---|---|---|---|---|
| EM | | | -10.02 | 0.52 | 0.63 | τ (days) | 191 (166 - 224) | 2 |
| EPM | | | -10.02 | 0.52 | 0.63 | τ (days) | 109 (95 - 195) | 4 |
| | | | | | | η (-) | 0.57 (0.52 - 0.96) | 12 |
| DM | | | -9.95 | 0.64 | 0.5 | τ (days) | 664 (490 - 760) | 11 |
| | -10.05 | 0.45 | | | | Dp (-) | 4.00 (2.80 - 3.93) | 8 |
| GM | | | -10.02 | 0.45 | 0.75 | τ (days) | 392 (296 - 1478) | 47 |
| | | | | | | α (-) | 0.70 (0.52 - 0.84) | 9 |
| TPLR | | | -10.02 | 0.45 | 0.76 | τf (days) | 42 (18 - 93) | 85 |
| | | | | | | τs (days) | 1623 (519 - 2638) | 29 |
| | | | | | | φ (-) | 0.29 (0.14 - 0.51) | 41 |

[a] σ = Standard deviation; KGE = Kling-Gupta Efficiency (Gupta et al., 2009). Statistical parameters of the simulated results correspond to the best-matching value of the objective function KGE.

[b] τ = tracer's mean transit time; η = parameter that indicates the ratio between the contribution of piston and exponential flow; Dp = dispersion parameter; τf and τs = transit time of fast and slow flows in biweeks; φ = flow partition parameter between fast and slow flow reservoirs. (-) = Dimensionless parameter. Uncertainty bounds (5-95 percentiles) of simulated parameters shown in parenthesis were estimated using the generalized likelihood uncertainty estimation (GLUE, Beven and Binley, 1992).

[c] MI = Measure of identification (Segura et al., 2012), i.e., ratio of behavioral parameter range to initial parameter range.





Table 6. Statistical parameters of observed and simulated δ¹⁸O for all catchments using the exponential model (EM).

| Catchment | Observed δ¹⁸O | | Simulated δ¹⁸O | | | Simulated MTT[b] | |
| --- | --- | --- | --- | --- | --- | --- | --- |
| | Mean | σ[a] | Mean | σ[a] | KGE[a] | (years) | (days) |
| | (‰) | (‰) | (‰) | (‰) | (-) | | |
| M1 | −10.63 | 0.37 | −10.51 | 0.44 | 0.48 | 0.54 (0.48 − 0.63) | 194 (171 − 227) |
| M2 | −10.46 | 0.53 | −10.51 | 0.63 | 0.61 | 0.43 (0.38 − 0.51) | 156 (137 − 183) |
| M3 | −10.64 | 0.23 | −10.51 | 0.26 | 0.48 | 0.73 (0.64 − 0.86) | 264 (232 − 310) |
| M4 | −10.63 | 0.27 | −10.51 | 0.3 | 0.48 | 0.67 (0.59 − 0.78) | 240 (212 − 280) |
| M5 | −10.51 | 0.39 | −10.51 | 0.46 | 0.53 | 0.52 (0.46 − 0.61) | 188 (165 − 219) |
| M6 | −10.37 | 0.42 | −10.43 | 0.5 | 0.59 | 0.52 (0.46 − 0.61) | 188 (164 − 220) |
| M7 | −8.93 | 2.92 | −10.02 | 2.93 | 0.84 | 0.15 (0.12 − 0.18) | 53 (45 − 64) |
| M8 | −10.05 | 0.45 | −10.02 | 0.52 | 0.63 | 0.53 (0.46 − 0.62) | 191 (167 − 224) |

[a] σ = Standard deviation; KGE = Kling-Gupta Efficiency. Statistical parameters of the simulated results correspond to the best-matching value of the objective function KGE.

[b] Uncertainty bounds (5-95 percentiles) of the simulated mean transit time (MTT) shown in parenthesis were estimated using the generalized likelihood uncertainty estimation (GLUE).





Table 7. Coefficient of determination ($R^2$) between the mean transit time (MTT) and i)
landscape features and ii) hydrological variables for each of the catchments. Catchments M3
and M4 (additional spring water source, see Figure 1) and M7 (at a flat hilltop disconnected
from the hillslopes) are not included in the regressions; except for electrical conductivity, i.e.,
all catchments are considered (Figure 9).

| Landscape features | | | Hydrologic variables | | |
|---|---|---|---|---|---|
| *Vegetation* | | | *General features* | | |
| Cushion plant | 0.29 | | Runoff coefficient | 0.62 | |
| Tussock grass | − 0.31 | | Total runoff | 0.29 | |
| | | | Precipitation | − 0.17 | |
| *Soil Type* | | | Average specific discharge | 0.21 | |
| Histosol | 0.13 | | | | |
| Andosol | − 0.13 | | *Streamflow rates* | | |
| | | | $Q_{99}$ | 0.42 | |
| *Geologic formation* | | | $Q_{90}$ | 0.18 | |
| Quimsacocha | 0.04 | | $Q_{80}$ | 0.06 | |
| Turi | 0.12 | | $Q_{70}$ | 0.09 | |
| Quaternary deposits | − 0.51 | | $Q_{60}$ | 0.06 | |
| | | | $Q_{50}$ | 0.01 | |
| | | | $Q_{40}$ | 0.10 | |
| *Topographic features* | | | $Q_{30}$ | − 0.02 | |
| Average slope | **− 0.78** | | $Q_{20}$ | − 0.14 | |
| Slope 0%−20% | **0.85** | | $Q_{10}$ | − 0.61 | |
| Slope 20%−40% | **− 0.90** | | $Q_5$ | − 0.62 | |
| Area | 0.13 | | | | |
| TWI | − 0.03 | | *Water intrinsic properties* | | |
| Flow path length (L) | 0.23 | | Electrical conductivity | **0.90** | |
| Flow path gradient (G) | − 0.02 | | | | |
| L/G | 0.23 | | | | |

Signs indicate positive (no sign) or negative (−) correlation between parameters.
Values in bold are statistically significant to a 95% level of confidence ($p < 0.05$).
[a] TWI = Topographic wetness index (Beven and Kirby, 1979).





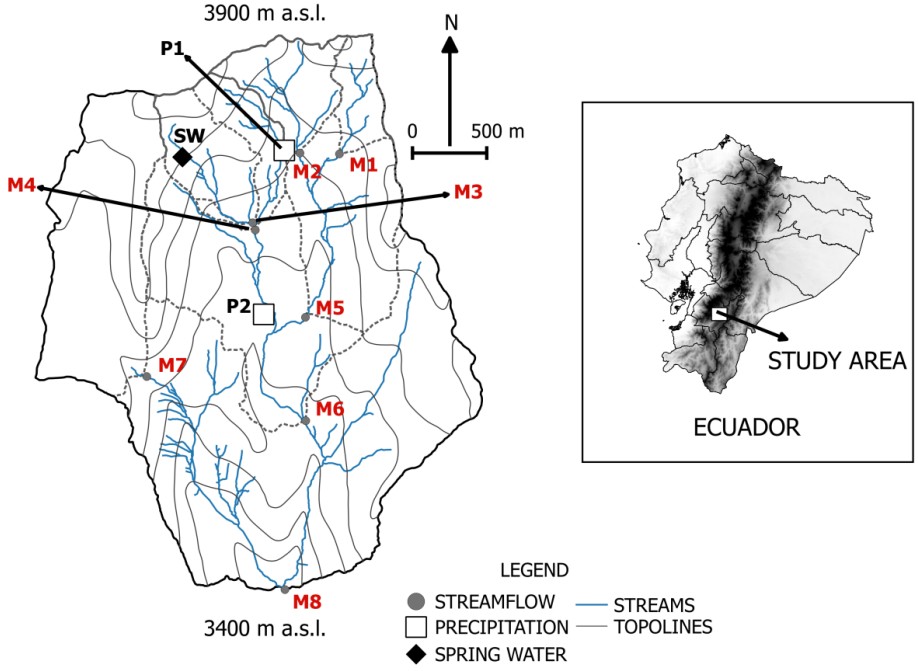

3    Figure 1. Location of the study area, and the isotopic monitoring stations in the Zhurucay

4    observatory for: Streamflow (M), and Precipitation (P). SW is a spring water source upstream

5    the outlet of catchments M3 and M4.

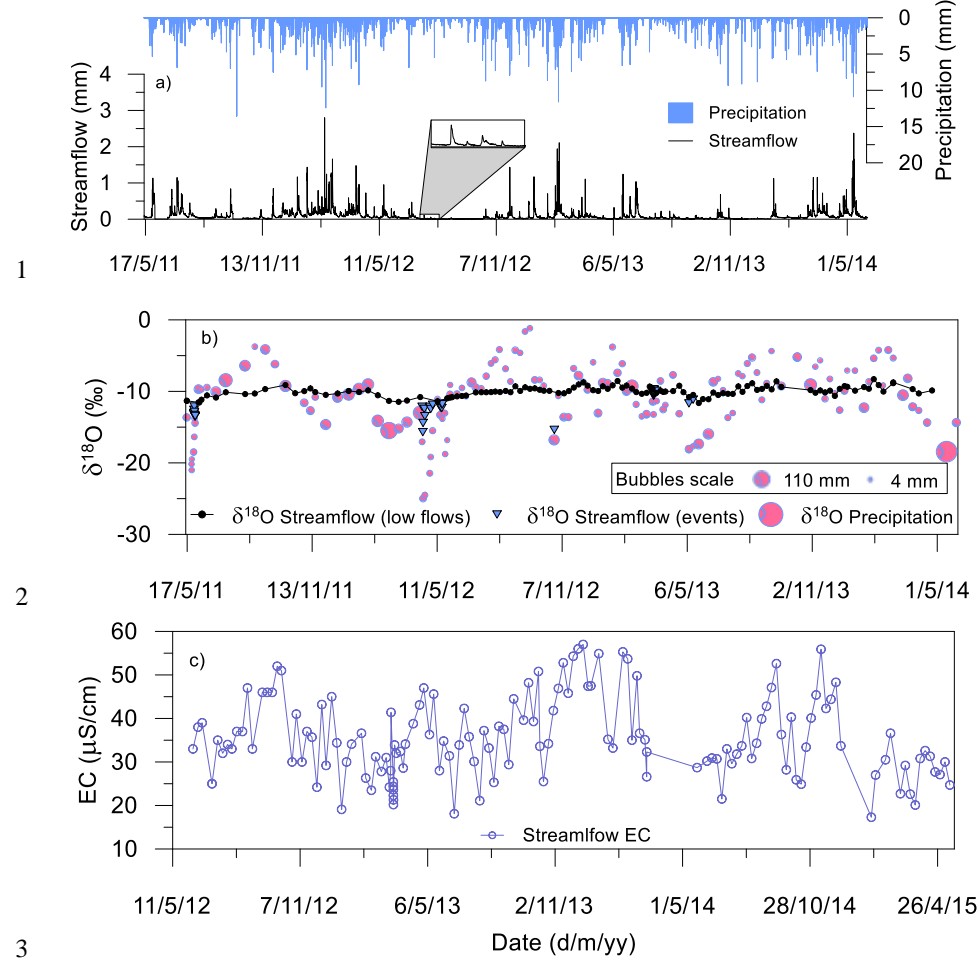

Figure 2. a) Hourly precipitation and unit area streamflow; b) δ18O isotopic composition in
precipitation and streamflow for 3 years (May 2011-May 2014); and c) electrical conductivity
for 3 years (May 2012-May 2015) at the catchment outlet (M8, see location in Figure 2). The
size of the bubbles in plot b) indicates the relative cumulative rainfall in millimeters for each
collected sample.




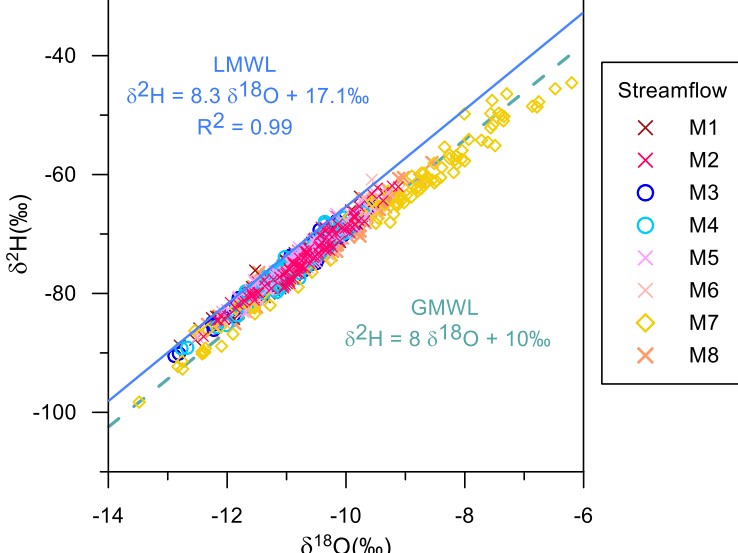

3  Figure 3. Relationship between the water stable isotopes ($\delta^2$H and $\delta^{18}$O) in streamflow for

4  water samples collected within the Zhurucay observatory. The Local Meteoric Water Line

5  (LMWL) and the Global Meteoric Water Line (GMWL) are also plotted.





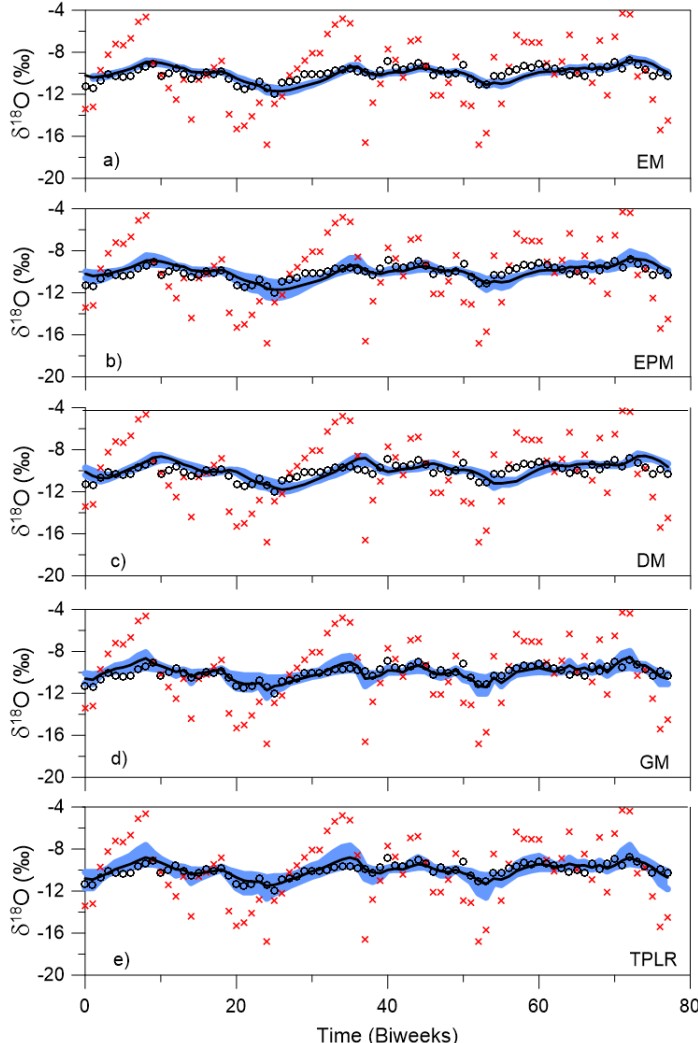

Figure 4. Fitted results of the five lumped parameter models used to simulate the temporal
variability in the δ18O streamflow composition at the outlet of the basin (M8). (a) Exponential
model (EM); (b) exponential-piston model (EPM); (c) dispersion model (DM); (d) gamma
model (GM); and (e) two parallel linear reservoir model (TPLR). The open circles represent
the observed isotopic composition in streamflow; the red crosses represent the isotopic



composition in precipitation; the black line represents the best simulated isotopic composition
in streamflow according to the KGE (Gupta et al., 2009) objective function; and the blue
shaded area corresponds to the 5-95% confidence limits of the possible solutions from the
parameter sets within the range of behavioral solutions, i.e., solutions which yield at least
95% KGE.





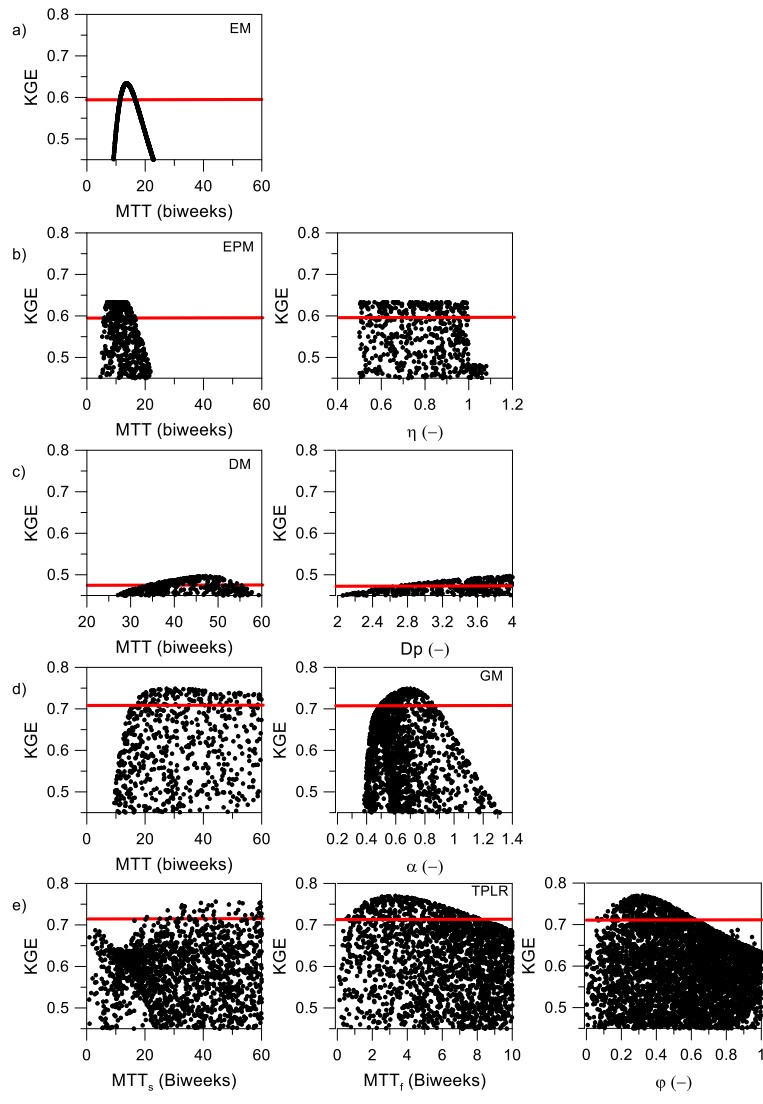

3    Figure 5. Monte Carlo simulation of the fitted parameters of the five lumped parameter

4    models used to simulate the $\delta^{18}O$ streamflow composition at the outlet of the basin (M8)

5    shown in figure 3. a) EM; b) EPM; c) DM; d) GM; and e) TPLR. The (-) symbol in the x-axes



denotes that fitting parameter is dimensionless. Horizontal red lines indicate threshold of
behavioral solutions (at least 0.95 of maximum KGE).





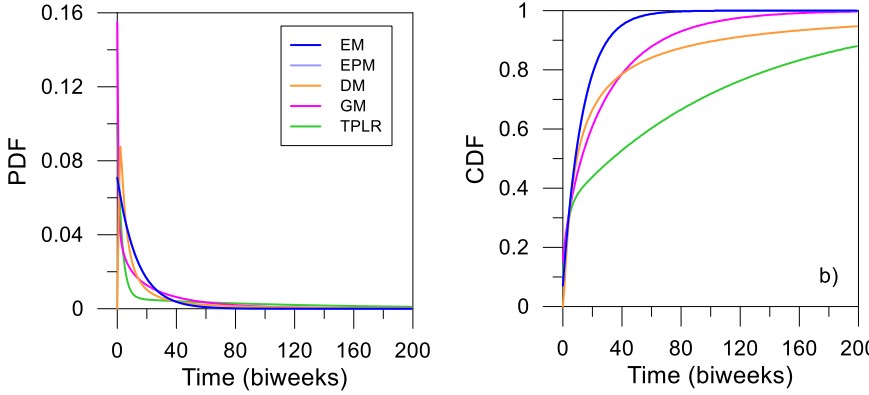

Figure 6. Probability (PDF) and cumulative (CDF) density functions for each transit time
distributions (TTD) used to simulate the $\delta^{18}$O streamflow composition at the outlet of the
basin (M8). TTDs correspond to the best-matching values of the objective function KGE.



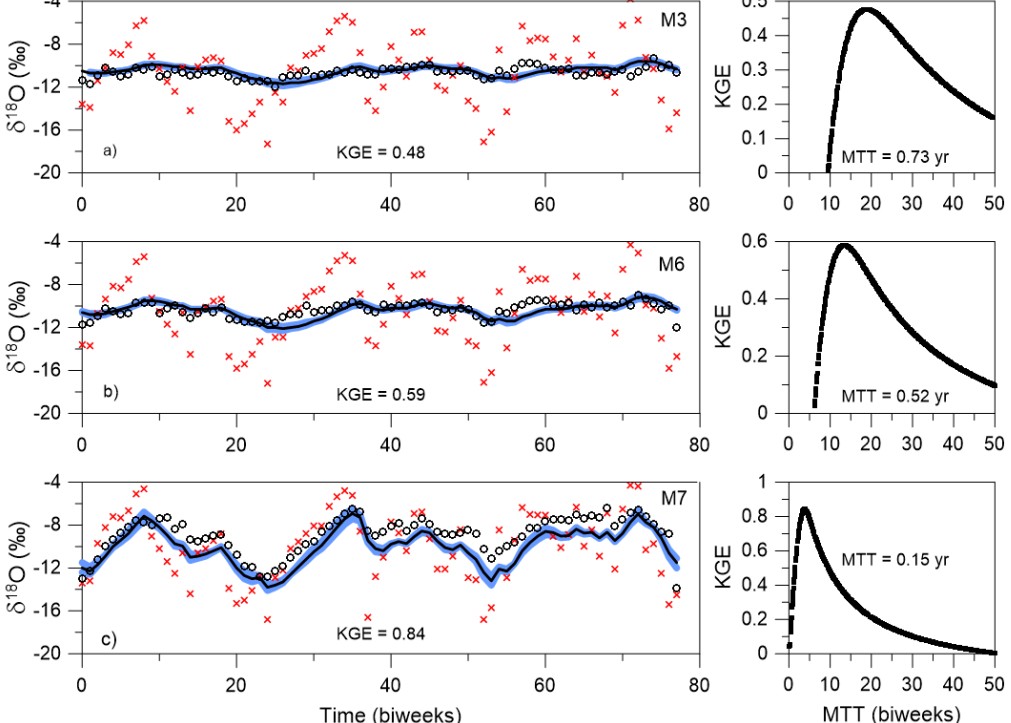

Figure 7. Fitted results and Monte Carlo simulations of the fitted parameters of the
exponential model (EM) used to simulate the $\delta^{18}O$ streamflow composition in the catchments:
a) M3; b) M6; and c) M7. The open circles represent the observed isotopic composition in
streamflow; the red crosses represent the isotopic composition in precipitation; the black line
represents the best simulated isotopic composition in streamflow according to the KGE
objective function; and the blue shaded area corresponds to the 5-95% confidence limits of
the possible solutions from the MTT fitting parameters within the range of behavioral
solutions, i.e., solutions which yield at least 95% KGE. Panels on the right represent the
explored parameter range for the MTT parameter and the KGEs associated to each of them.



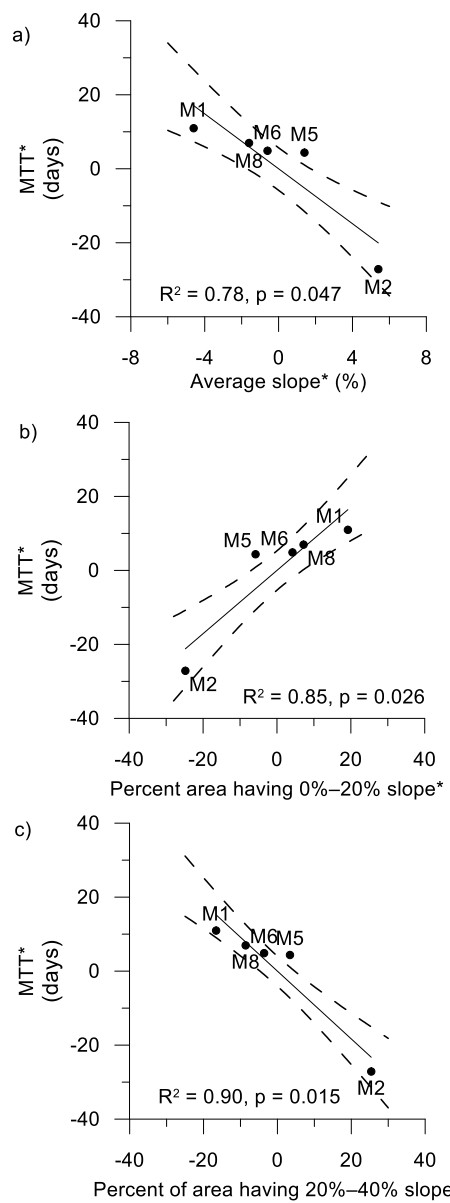

3    Figure 8. Correlations between mean transit time (MTT) and topographic indexes of the

4    catchments: a) average catchment slope; b) catchment area with slopes between 0% and 20%;



and c) catchment area with slopes between 20% and 40%. Catchments M3 and M4 (additional
spring water source, see Figure 1) and M7 (at a flat hilltop disconnected from the hillslopes)
are not included in the regressions. Solid lines are linear regressions and dashed lines are the
90% confidence intervals of the regressions. * Indicates parameters are normalized by their
mean




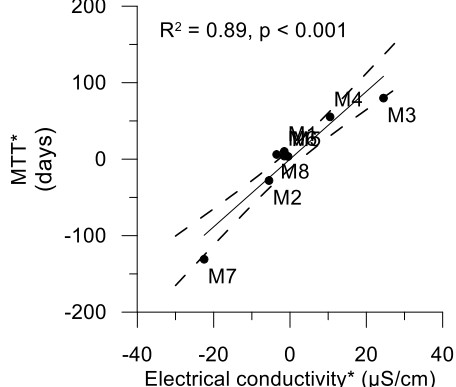

Figure 9. Correlation between mean transit time (MTT) and mean electrical conductivity for
weekly measurements of stream water samples collected during three years (June 2012-June
2015). Solid line is the linear regression and the dashed lines are the 90% confidence intervals
of the regression. * Indicates parameters are normalized by their mean.