# Peer review of "Insights on the water mean transit time in a high-elevation tropical ecosystem"

_Hydrology and Earth System Sciences, 2015_

## Referee Comment (RC1) · Anonymous Referee #1 · 19 Feb 2016

**GENERAL COMMENTS**

The paper attempts to explore transit time distributions (TTD) in a high-elevation tropical ecosystem by using a detailed hydrologic and isotopic record from eight nested catchments located in southern Ecuador. Although the data are extremely interesting and unique in quality and location, the transit time analysis is performed through a method (the lumped convolution approach) which is likely to include an aggregation bias, especially for systems with a high degree of heterogeneity and non-stationarity (see the recent papers by Kirchner, [2016a,b]). In simple terms, even if the transfer function approach allows a fair simulation of the measured isotopic signal, the system mean transit time is not necessarily realistic, due to the structural uncertainties in the quantification of the older water components. This emerges in Figures 4 and 6, where different TTD (with different MTT) result in similar model performances. Moreover, the

paper ignores the recent advances in hydrologic transport and TTD (see the list of suggested literature), which are now widespread within the hydrologic community and have clarified the concept of TTD in the light of non-stationarity. The manuscript is clear, well written and easy to follow, but the methods pose some serious concern on the paper's conclusions.

**DETAILED COMMENTS**

Page 7, line 18: the authors say that kinetic fractionation by evaporation can be neglected, however looking at Figure 3 it seems that the majority of stream water samples plot below the LMWL. How can this behavior be explained?

P. 9, I. 3: the variable tau in Eq. (1) and (2) is not the mean transit time. It is just the dummy variable in the integral, which spans the transit time domain [0, +inf].

P. 10, I. 22-26: I did not get why the model is run twice to get the behavioral set of parameters.

P. 10, I. 28: the MI index seems to be very arbitrary depending on the choice of the prior parameter distribution. Segura et al., [2012] provide a partial explanation for their choice of the prior, which is here missing.

P. 13, I. 27: the terminology "MTT probability density function" seems to refer to the pdf of MTT obtained from the posterior parameter distribution.

P. 14, I. 1-13: this is to me a clear example of the indetermination of the MTT. Different parameterizations of the TTD are able to provide good, similar simulations of the isotopic signal, but result in rather different MTT. While it is reasonable to choose a model because its parameters are more constrained in the simulation of a specific target, this does not allow to extrapolate that its MTT is the "right" one.

P. 15, I. 21: what is meant by "completely" recovered? Is there a threshold (e.g. 99%) on the recovered mass?

SUGGESTED LITERATURE

Kirchner, J. W. (2016a). Aggregation in environmental systems - Part 1: Seasonal tracer cycles quantify young water fractions, but not mean transit times, in spatially heterogeneous catchments. Hydrology and Earth System Sciences, 20(1), 279–297. http://doi.org/10.5194/hess-20-279-2016

Kirchner, J. W. (2016b). Aggregation in environmental systems - Part 2: Catchment mean transit times and young water fractions under hydrologic nonstationarity. Hydrology and Earth System Sciences, 20(1), 299–328. http://doi.org/10.5194/hess-20-299-2016

Botter, G., Bertuzzo, E., & Rinaldo, A. (2011). Catchment residence and travel time distributions: The master equation. Geophysical Research Letters, 38(11). http://doi.org/10.1029/2011GL047666

van der Velde, Y., Torfs, P. J. J. F., van der Zee, S. E. A. T. M., & Uijlenhoet, R. (2012). Quantifying catchment-scale mixing and its effect on timevarying travel time distributions. Water Resources Research, 48(6), W06536. http://doi.org/10.1029/2011WR011310

McMillan, H., Tetzlaff, D., Clark, M., & Soulsby, C. (2012). Do time-variable tracers aid the evaluation of hydrological model structure? A multimodel approach. Water Resources Research, 48(5). http://doi.org/10.1029/2011WR011688

Heidbüchel, I., Troch, P. a., Lyon, S. W., & Weiler, M. (2012). The master transit time distribution of variable flow systems. Water Resources Research, 48(6), W06520. http://doi.org/10.1029/2011WR011293

Davies, J., Beven, K., Rodhe, A., Nyberg, L., & Bishop, K. (2013). Integrated modeling of flow and residence times at the catchment scale with multiple interacting pathways, 49, 4738–4750. http://doi.org/10.1002/wrcr.20377

Bertuzzo, E., Thomet, M., Botter, G., & Rinaldo, A. (2013). Catchment-scale herbi-

СЗ

cides transport: Theory and application. Advances in Water Resources, 52, 232–242. http://doi.org/10.1016/j.advwatres.2012.11.007

Hrachowitz, M., Savenije, H., Bogaard, T. a., Tetzlaff, D., & Soulsby, C. (2013). What can flux tracking teach us about water age distribution patterns and their temporal dynamics? Hydrology and Earth System Sciences, 17(2), 533–564. http://doi.org/10.5194/hess-17-533-2013

Harman, C., & Kim, M. (2014). An efficient tracer test for time-variable transit time distributions in periodic hydrodynamic systems. Geophysical Research Letters, 41(5), 1567–1575. http://doi.org/10.1002/2013GL058980

Soulsby, C., Birkel, C., & Tetzlaff, D. (2014). Assessing urbanization impacts on catchment transit times. Geophysical Research Letters, 41(2), 442–448. http://doi.org/10.1002/2013GL058716

van der Velde, Y., Heidbüchel, I., Lyon, S. W., Nyberg, L., Rodhe, A., Bishop, K., & Troch, P. a. (2014). Consequences of mixing assumptions for time-variable travel time distributions. Hydrological Processes. http://doi.org/10.1002/hyp.10372

Harman, C. J. (2015). Time-variable transit time distributions and transport: Theory and application to storage-dependent transport of chloride in a watershed. Water Resources Research, 51(1), 1–30. http://doi.org/10.1002/2014WR015707

Klaus, J., Chun, K. P., McGuire, K. J., & McDonnell, J. J. (2015). Temporal dynamics of catchment transit times from stable isotope data. Water Resources Research, 51(6), 4208–4223. http://doi.org/10.1002/2014WR016247

Benettin, P., Kirchner, J. W., Rinaldo, A., & Botter, G. (2015). Modeling chloride transport using travel time distributions at Plynlimon, Wales. Water Resources Research, 51(5), 3259–3276. http://doi.org/10.1002/2014WR016600

Hrachowitz, M., Fovet, O., Ruiz, L., & Savenije, H. H. G. (2015). Transit time distributions, legacy contamination and variability in biogeochemical 1/f  $\alpha$  scaling: how are

hydrological response dynamics linked to water quality at the catchment scale? Hydrological Processes, 29(25), 5241–5256. http://doi.org/10.1002/hyp.10546

Birkel, C., Soulsby, C., & Tetzlaff, D. (2015). Conceptual modelling to assess how the interplay of hydrological connectivity, catchment storage and tracer dynamics controls nonstationary water age estimates. Hydrological Processes, 29(13), 2956–2969. http://doi.org/10.1002/hyp.10414

Soulsby, C., Birkel, C., Geris, J., Dick, J., Tunaley, C., & Tetzlaff, D. (2015). Stream water age distributions controlled by storage dynamics and nonlinear hydrologic connectivity: Modeling with high-resolution isotope data. Water Resources Research, 51(9), 7759–7776. http://doi.org/10.1002/2015WR017888

Benettin, P., Bailey, S. W., Campbell, J. L., Green, M. B., Rinaldo, A., Likens, G. E., McGuire, K. J., & Botter, G. (2015). Linking water age and solute dynamics in streamflow at the Hubbard Brook Experimental Forest, NH, USA. Water Resources Research, 51(11), 9256–9272. http://doi.org/10.1002/2015WR017552

---

## Referee Comment (RC2) · C. Birkel (Referee) · 27 Feb 2016

The manuscript "Insights on the water mean transit time in a high-elevation tropical ecosystem" by Mosquera et al. under review in Hydrol. Earth Syst. Sci. Discuss., doi:10.5194/hess-2015-546, 2016 presents an attempt to investigate MTTs of a nested paramo catchment system in Ecuador with the purpose to tease out dominant controls on water transit time. The authors were able to identify relatively short transit times (< 1yr) compared to other environments in different climatic regions. The MTTs in their study site are mainly controlled by the catchment slope in relation to the dominant wetland soils. The experimentally derived dataset for this tropical ecosystem is unique and interesting to the HESS readership and beyond. The analysis is mostly sound and the paper generally well-written and structured.

Having said that, the paper struggles in parts to clearly convey the main points in line with the objectives of the study and could be shortened. I am missing a discussion around arguments that the MTT is not a meaningful catchment descriptor and the recent tendency towards the recognition of the time-variant nature of transit times. I do think that there are merits in using the MTT to characterize catchment systems particularly considering the constraints and limitations working in tropical environments; it should, however, be more clearly argued. Furthermore, there are some model decisions that should be more clearly explained, which also likely leads to additional analysis strengthening the paper and its line of arguments. Nevertheless, I think this is nothing that cannot be fixed with a careful revision to improve clarity and focus of the paper and I therefore support publication of this paper with some revisions.

Specific comments:
Abstract:
Line 21: I'm not sure if the paper is about streamwater MTT as you excluded high-flow events from the analysis.

Key words:
Line 15: I suggest to simplify and reduce the key words to attract more online search results, e.g.: Ecohydrology, MTT, runoff generation, Andean paramo, Histosols, Ecuador.

Introduction:

Page 3, Line 2: This is true for Latin America, but there are a few more studies in the tropics. You could even refer to Muñoz-Villers and McDonnell (2012) in this context.

Page 3, Line 17: I will come back to this point, but I think it's very likely that there's also a considerable near-surface runoff component as seen in other environments (you refer to Scotland and Sweden below) with organic rich wetland soils that remain saturated for much of the year. I, however, don't know the paper in review you cite here.

Page 3, Line 29: This isn't entirely true, I'm afraid, because the dominating runoff generation process based on various tracer studies is a rapid near-surface flow. The subsurface component is a deeper and slower groundwater flux. Therefore, the wetland contribution can be quantified very well in form of near-surface saturation overland flow.

Page 4, Line 9: I think Broxton et al. (2009) worked in Arizona, USA. You could also specify the control you are referring to as in this case it was "aspect".

Study site:

Page 5, Line 8: This is an awkward sentence, please revise.

Line 10: seasonality, primarily.

Line 23: please, spell out INV.

Page 6, Line 4-5: Please, revise this sentence.

Line 27: Please, indicate model and make of the equipment.

Methods:

Page 9, Line 26: ...is based...

Line 27: I'm not sure I follow the second point.

Page 10, Line 6: In this case, I suggest to consistently refer to a baseflow MTT and not streamwater MTT.

Page 11, Line 2-5: I fully agree that you seek to identify the best-performing and most parsimonious model. However, you don't really compare the models using a criterion for model selection (e.g., AIC, BIC or adjusted R2) that penalizes the number of parameters in combination with a goodness-of-fit measure. The MI criterion looks at how identifiable one parameter is, but not at the combined effect of more than one parameter used to calibrate the model.

Page 11: How were models generated? Using a uniformly sampled Monte Carlo procedure?

Line 16: mainly?

Results:

Page 12, Line 12: Runoff coefficients show...

Page 13, Line 4-26: I'm not convinced by some of the statements present in this paragraph. For example, the best-fit gamma model compared to the best-fit exponential model does show a quite significant increase in performance (from 0.63 to 0.75) that can justify the use of one additional fitting parameter. On the other hand, a third fitting parameter resulted in an increased performance of only 0.01. The poorest model seems to be the DM with a best-fit of KGE=0.5. Based on this, one could qualitatively reject the DM and TPLR models as suitable models compared to the EM and GM. However, the decision between the EM and GM models should be informed by a model selection criterion such as the AIC (see comment above) that evaluates the combined effect of the parameters on model performance.

Page 14, Line 12: Please, revise this sentence.

Page 14: I think that large parts here could be moved into the discussion or simply be deleted as later sections pick up on these issues. This would allow to shorten the m/s focussing on presenting the key results and later discussion in the light of the wider literature.

Page 15, Line 26: just use MTT

Page 16, Line 5-12: Please, separate this very long sentence into smaller parts.

Discussion:

Page 17, Line 26: more depleted?

Page 18, Line 1: I think it would be better to indicate that baseflow MTT was analysed.

Line 3: identifiability?

Line 20: Was TMCF previously defined?

Line 30: remain.

Page 19, Line 10: You somehow have to convince me that this actually is subsurface stormflow. I haven't seen the in review paper you mention in this context and all the

evidence you show tells me that the dominating runoff generation mechanism is near-surface saturation overland flow due to little mixing with deeper soil horizons, short MTTs, etc.

Line 20: You previously said up to 2.2 years in this context.

Page 20, Line 6: I think Hrachowitz et al. (2009) argued with saturation overland flow.

Line 11: solutes.

Page 22, Line 1: explain.

Line 8: Please, revise this sentence.

Line 14: Isn't this simply the slope?

Line 24: I find the "regulation capacity" is coming a bit out of nowhere. What exactly do you mean by this? Is it in the sense of resilience or simply that the turn-over is quick and what goes in comes out with little delay?

Page 23, Line 2: It's the first time that you mention that SOF wasn't previously observed in the study catchment. This information needs to come earlier. I also think this whole paragraph can be shortened towards the key messages presented at the very end.

Line 29: Please, revise this sentence.

Tables:

Shouldn't the current Table 2 come before you present the models (Table 1)?

Current Table 1: I'm a bit confused about some decisions concerning the choice of initial parameter intervals. Why was the upper limit of tau set at 200 biweeks? This makes 2800days and over 7 yrs of TT, something stable isotopes aren't able to detect anyways (Stewart et al., 2010). Further, why was the lower limit of beta (GM) set to 0.5? In the case of low TT this could be well below 0.5 and on a global scale the average resulted to be at around 0.5 (Godsey et al., 2009). With the current lower limit in place you potentially miss suitable parameters that would also result in lower MTTs compared to current best-fit results; an argument you used to reject the GM. Also, it seems odd to me that you don't report the parameter interval for beta as this is the parameter you calibrate. The MTT (tau) is only the result of beta*alpha.

Table 5: Similar issue here with the GM. I suggest to report the parameters alpha and beta.

Table 7: R2-values of 0.62 did not result significant? However, there's a relationship with flow characteristics particularly for the extremes and the runoff coefficient does seem to explain some of the spatial variability among catchments.

Figures:

Figure 2: What's the purpose of the streamflow inlet box? Could you not just show a log-scale to emphasize the low flow periods? Those event samples do show quite a bit of response to rainfall. What's the effect of pooling these out? Quite a bit shorter MTTs? Please, consider adjusting the different EC sampling period for comparison purposes.

Figure 5: Please, clarify if sampling was started below alpha = 0.5 (GM) contrary to the information from Table 1. Again, I suggest to present the parameters alpha and beta.

Figure 6: Is EPM missing in the right panel?

Figure 8: If the MTT is normalized shouldn't it be unitless?

References I used:

Godsey, S. E., Aas, W., Clair, T. A., de Wit, H. A., Fernandez, I. J., Kahl, J. S., Malcolm, I. A., Neal, C., Neal, M., Nelson, S. J., Norton, S. A., Palucis, M. C., Skjelkvåle, B. L., Soulsby, C., Tetzlaff, D. and Kirchner, J. W. (2010), Generality of fractal 1/f scaling in catchment tracer time series, and its implications for catchment travel time distributions. Hydrol. Process., 24: 1660–1671. doi: 10.1002/hyp.7677.

Stewart MK, Morgenstern U, McDonnell JJ. 2010. Truncation of stream residence time: How the use of stable isotopes has skewed our concept of streamwater age and origin. *Hydrological Processes* **24**: 1646–1659.

---

## Author Comment (AC1) · 29 Mar 2016

Referee 1:

Reply: We really appreciate the review provided by referee 1 (R1) and are glad that our work gives rise to quite interesting discussion of catchment heterogeneity and non-stationarity conditions in the context of robust MTT estimations. R1 comments focus on four main aspects: 1) site conditions, 2) aggregation bias, 3) model evaluation and performance, and 4) selected methodology. Hence, we provide responses to each of them separately below.

R1: The paper attempts to explore transit time distributions (TTD) in a high-elevation tropical ecosystem by using a detailed hydrologic and isotopic record from eight nested catchments located in southern Ecuador.

Reply: First, we want to emphasize that our system is characterized by unique hydrometeorological and landscape characteristics in comparison to other systems: i) mean annual precipitation, runoff, and evapotranspiration are similar across the entire catchment ($1284\pm18$ mm yr-1, $788\pm54$ mm yr-1, $496\pm61$ mm yr-1, respectively) (Mosquera et al., 2015), ii) precipitation is evenly distributed year-round with very low degree of seasonality (Padrón et al., 2015), iii) isotopic fractionation by evaporative effects is virtually negligible (Mosquera et al., 2016) as a result of the year-round high relative humidity ($\sim$90%) (Córdova et al., 2015), iv) the soils are shallow and poorly developed across the entire catchment ($\sim$1m deep), and v) the geology is relatively young and homogeneous (Coltorti and Ollier, 2000). We believe our system presents a high degree of homogeneity across the entire basin as a result of the mentioned landscape configuration and local hydrometeorological conditions. A description of these conditions will be included in the final version of the manuscript.

R1: Although the data are extremely interesting and unique in quality and location, the transit time analysis is performed through a method (the lumped convolution approach) which is likely to include an aggregation bias, especially for systems with a high degree of heterogeneity and non-stationarity (see the recent papers by Kirchner, [2016a,b]).

Reply: We appreciate the comment about how extremely interesting and unique our data are. Regarding our approach, we will first focus on the issue of heterogeneity. It is certainly true that heterogeneity is "a fundamental problem" in the investigation of catchment behavior, because the scale of investigation influences the type of hydrological processes that can be identified. This issue is very well captured in Figure 4 of Kirchner (2016a) which exemplifies how MTT estimations can be affected as a result of aggregation across scales in heterogeneous catchments. We recognized this issue since our study design by considering a nested monitoring configuration. This configuration allowed us to investigate the variability of hydrological processes across scales and to characterize the system's degree of heterogeneity. As mentioned above, the characteristics of the Zhurucay basin, provide quasi-homogeneous conditions or

low degree of heterogeneity, which most likely significantly reduces the issue of aggregation bias in MTT estimations. The landscape homogeneity in our system is evident considering that the same TTD represent the subsurface hydrologic system's behavior at all catchments, with a relatively small range of variation of the estimated MTTs (0.51±0.17 yr). Thus, we consider that given the homogeneous catchment characteristics and small seasonality of local environmental conditions in our system, applying the LCA to our water stable isotopic dataset is a first step over which to build-up improved catchment functioning understanding. Regarding the nonstationarity issue (Kirchner, 2016b), we first want to clarify that using a certain methodology does not imply that other ones are ignored. The LCA is an approach that assumes steady-state conditions in the system, assumption acknowledged in our paper. However, as highlighted in manuscript title (i.e., "Insights"), our study aims to set a baseline for the application of modeling techniques using water stable isotopes in tropical ecosystems above the tree line, in which in general, there is very scarce hydrologic information. To our knowledge, this is the first contribution regarding the modeling of MTT in páramo ecosystems. Although recent advances in hydrologic research, such as the ones listed by R1, have provided theoretical evidence of the importance of recognizing the unsteady nature of hydrologic processes highlighting the possible shortcomings of the LCA, our paper is not a methodological contribution regarding MTT modeling theory but rather about understanding catchment functioning in a high-elevation tropical region. Indeed one of our future goals is to apply modeling techniques that explicitly recognize nonstationarity and storage dynamics in the hydrological behavior of this tropical ecosystem (e.g.,Birkel et al., 2015; Harman, 2015; Hrachowitz et al., 2013). These analyses, however, are beyond the scope of this paper. As mentioned above, our intention is to generate hydrologic knowledge in this understudied region. That said, we also believe our data set is valuable because it provides a concrete example to test the applicability, limitations, and constraints of different MTT modeling methodologies. In fact, based on the homogeneous and uniform hydrometeorological characteristics of our site, in contrast to many of the temperate regions in which time-variant MTT modeling has

been developed, we anticipate our dataset and the results from this study to allow for benchmark testing of MTT methodologies in regions with low climate seasonality. At the same time, we will acknowledge the growing recognition of the time-variant nature of transit times and make sure to reference and highlight the value of such modeling methodologies and recent findings related to them. It is relevant to note a recent application of conceptual modeling for the investigation of non-stationary conditions in a wet Scottish upland catchment where runoff generation processes mainly occur in the riparian Histosol soils with high storage capacity (Birkel et al., 2015), as in our study site. These authors detected non-stationary characteristics in water age distributions only during extreme weather conditions (extensive dry or wet periods) and attributed this behavior to the large mixing capacity of the Histosol soils, "which acts as an isostat moderating isotope variability and limiting the time variance of water age". The latter has been clearly observed in the Zhurucay basin, where the isotopic composition of the organic horizon of the Histosol soils remains virtually constant and matches the isotopic composition in the streams year-round (Mosquera et al., 2016). This, in combination with the nearly uniform climate characteristics in our site, supports the utility of steady-state approaches in our system.

R1: In simple terms, even if the transfer function approach allows a fair simulation of the measured isotopic signal, the system mean transit time is not necessarily realistic, due to the structural uncertainties in the quantification of the older water components. This emerges in Figures 4 and 6, where different TTD (with different MTT) result in similar model performances.

Reply: With regards to the uncertainties related to the quantification of old water components, we consider this is not a significant issue in our system. Evidence of very low (insignificant) "old" water contributions (i.e., deep groundwater contributions to discharge) in the Zhurucay basin has been found by Crespo et al. (2011) and Mosquera et al. (2016) and by Buytaert and Beven (2011) in a nearby páramo catchment in South Ecuador. It appears that this results from the combination of relatively young

and homogeneous geology with the high storage capacity of the porous organic horizon of the páramo soils in combination with their low level of development (soils are generally less than 1 m deep). Results from our study support this interpretation. This is evidenced by the fact that the two TTDs functions (Gamma and two parallel linear reservoir) that incorporate an "old" water component yield parameter values that are not well constrained. Instead the exponential model provided a robust representation with a clearly defined parameter. In this sense, it must be highlighted that our procedure to identify the TTD that best describes our system hydrologic behavior did not only take into account the goodness-of-fit of the objective function but also the level of identification of the function parameters and a process-based interpretation of the results (see below). Regarding the similar performance of the model using different TTDs, Timbe et al. (2014) conducted a detailed analysis of the uncertainties related to the use of different TTDs in MTT modeling using the LCA. They also found that several TTDs provide high goodness-of-fit between predictions and observations, but poor parameter identifiability for some TTDs calibrated parameters, as has also been observed by other researchers (e.g., Hrachowitz et al., 2009). As such they recommend that for achieving meaningful MTT estimates from the LCA, it is at least needed to: 1) used several TTDs, 2) evaluate predictions uncertainty, and 3) assess parameter identifiability for each TTD function. Following these recommendations, we conducted an assessment of the performance of different TTDs, considering not only the best fit but also the uncertainty of the predictions, the parameter identifiability, and a process-based interpretation in light of the detailed hydrometric, isotopic, and biophysical landscape information which has been collected at our study site over the last five years (Córdova et al., 2015; Mosquera et al., 2015, 2016; Padrón et al., 2015; Quichimbo et al., 2012) to select the model that best describes the hydrologic functioning of the system. In addition, we will also include an analysis using a metric for model selection (Akaike information criterion, AIC). This analysis has confirmed the EM as the one that best describes the hydrologic functioning in our system.

R1: Moreover, the paper ignores the recent advances in hydrologic transport and TTD

(see the list of suggested literature), which are now widespread within the hydrologic community and have clarified the concept of TTD in the light of non-stationarity. The manuscript is clear, well written and easy to follow, but the methods pose some serious concern on the paper's conclusions.

Reply: We appreciate this comment. We will include in the paper a clear discussion about the recent time variant advances in hydrologic modeling. However as mentioned above, given the high degree of homogeneity of our system we believe that assuming steady-state conditions is justified as a first step over which to build-up improved catchment functioning understanding using hydrometric-tracer based hydrologic modeling in this understudied region. In addition we believe that limiting transit time modeling efforts only to recent methodologies prevents us to gain knowledge from different information provided by different approaches that ultimately, altogether, can help improve catchment functioning understanding by fulfilling/complementing information yielded by each of them. This is particularly critical in the case of MTT modeling under non-stationary conditions, an approach that is currently under development, and as result, there is yet no unified methodology that can be globally applied. Indeed there are very few applications of such methodology, most of which have yield results with high degree of uncertainty (Harman, 2015; Klaus et al., 2015; McMillan et al., 2012) or it is not even estimated (Davies et al., 2013; Heidbüchel et al., 2012; van der Velde et al., 2015), mainly as a result of expensive computation costs or high uncertainties related to the spatial variability of the input hydrometric and tracer field measurements. It is clear however, that given the mathematical limitations (Duvert et al., 2016; Seeger and Weiler, 2014), high-temporal resolution of tracer data required (Harman, 2015; Heidbüchel et al., 2012), and general unavailability of long-term tracer records (Hrachowitz et al., 2010; Klaus et al., 2015) also required for hydrological modeling under non-stationary conditions, the LCA is still a useful metric of storage and catchment functioning not only in understudied regions such as the tropics (e.g., Farrick and Branfireun, 2015; Muñoz-Villers et al., 2015; Timbe et al., 2014) but also elsewhere (e.g., Duvert et al., 2016; Hale and McDonnell, 2016; Hale et al., 2016; Hu

et al., 2015; Seeger and Weiler, 2014). In this sense, we agree with Christian Birkel (referee 2, R2) comment: "there are merits in using the MTT to characterize catchment systems particularly considering the constraints and limitations working in tropical environments" and are convinced that this "experimentally derived dataset for this tropical ecosystem is unique and interesting to the HESS readership and beyond", particularly taking into account the system's particular characteristics. We believe this contribution will become a benchmark study over which to build-up further hydrological processes understanding not only in this remote understudied region, but also more generally, in regions with low climate seasonality and catchments with low degree of heterogeneity. Future efforts will built upon the monitoring infrastructure and datasets that continue to be collected in the Zhurucay River Ecohydrological Observatory, which will allow for continual improvement in hydrologic interpretation by eventually incorporating some alternative modeling techniques (e.g., Birkel et al., 2015; Harman, 2015; Hrachowitz et al., 2013).

DETAILED COMMENTS

Page 7, line 18: the authors say that kinetic fractionation by evaporation can be neglected, however looking at Figure 3 it seems that the majority of stream water samples plot below the LMWL. How can this behavior be explained?

Reply: The isotopic composition in stream waters corresponds to a mixture of rainfall waters which have resided different periods within the system. Therefore, for streams and soil waters it is not feasible to directly evaluate the existence of evaporation effect on fractionation using a relationship like the LMWL, useful for precipitation, which purely depends on the atmospheric conditions at the source of water vapor and further at the study site. Fractionation effects in stream waters due to evaporation can be assessed through the d-excess of individual water samples, with samples with d-excess values falling 5‰ or more below that of the LMWL indicating evaporation (Brooks et al., 2012). At the study site, Mosquera et al. (2016) found that the d-excess of stream waters for all individual samples were above 5‰ (average 11.56±0.96‰. These authors

attributed this to the high year-round relative humidity (∼90%) in the Zhurucay basin (Córdova et al., 2015). This explains why kinetic fractionation in streamflow water samples can be neglected. We will include a discussion about kinetic fractionation and will reference the work of Mosquera et al. (2016) in the final version of the manuscript

P. 9, l. 3: the variable tau in Eq. (1) and (2) is not the mean transit time. It is just the dummy variable in the integral, which spans the transit time domain [0, +inf].

Reply: Thank you catching this typo. We will correct this in the final version of the manuscript.

P. 10, l. 22-26: I did not get why the model is run twice to get the behavioral set of parameters.

Reply: We first run the model 10,000 times considering a wide range of parameter values. This provided and idea about the range of acceptable values. Based on the latter, we narrowed the parameter space and run the model again until 1,000 solutions or more, corresponding to at least 95% of the KGE objective function (i.e., at least 1,000 behavioral solutions), were obtained. These 1000 solutions allowed strong identification of the 90% confidence interval using the GLUE methodology. This procedure will be clarified in the final version of the manuscript.

P. 10, l. 28: the MI index seems to be very arbitrary depending on the choice of the prior parameter distribution. Segura et al., [2012] provide a partial explanation for their choice of the prior, which is here missing.

Reply: We appreciate this observation. Our initial choice of parameters ranges was selected based on the work of Timbe et al. (2014) in their analysis of uncertainties related to the use of the LCA for MTT estimations. We will clarify this in the manuscript.

P. 13, l. 27: the terminology "MTT probability density function" seems to refer to the pdf of MTT obtained from the posterior parameter distribution.

Reply: You are correct. The pdf and cdf we referred to in the text correspond to the

distributions described based on the fitted parameter distributions. We will clarify this in the text.

P. 14, l. 1-13: this is to me a clear example of the indetermination of the MTT. Different parameterizations of the TTD are able to provide good, similar simulations of the isotopic signal, but result in rather different MTT. While it is reasonable to choose a model because its parameters are more constrained in the simulation of a specific target, this does not allow to extrapolate that its MTT is the "right" one.

Reply: This issue is been already discussed above in the general comment and we just want to emphasize that we carefully considered the uncertainty of the predictions and the parameter identifiability, in addition to the results of the simulation of the isotopic signal against the objective function and will further include the results from the AIC model selection metric that support that the EM is the one that best describes our system, as suggested by R2. Additionally, we conducted a process-based interpretation of these results in light of the detailed hydrometric, isotopic, and biophysical landscape information which has been collected at the Zhurucay basin over the last five years (Córdova et al., 2015; Mosquera et al., 2015, 2016; Padrón et al., 2015; Quichimbo et al., 2012) for selecting the TTD that best suites the hydrologic conditions of the system.

P. 15, l. 21: what is meant by "completely" recovered? Is there a threshold (e.g. 99%) on the recovered mass?

Reply: This means that if the tracer would have been injected as a single pulse, how much it would take to completely leave the system. In effect, certain proportion of the total injection will be recovered at a certain time after the injection, For example, Figure 6 depicts that for the EM 80% of the tracer is recovered at around 20 biweeks. Analogous analysis have been reported by Hrachowitz et al. (2009) and McGuire et al. (2005).

SUGGESTED LITERATURE

Kirchner, J. W. (2016a). Aggregation in environmental systems - Part 1: Seasonal tracer cycles quantify young water fractions, but not mean transit times, in spatially heterogeneous catchments. Hydrology and Earth System Sciences, 20(1), 279–297. http://doi.org/10.5194/hess-20-279-2016

Kirchner, J. W. (2016b). Aggregation in environmental systems - Part 2: Catchment mean transit times and young water fractions under hydrologic nonstationarity. Hydrology and Earth System Sciences, 20(1), 299–328. http://doi.org/10.5194/hess-20-299-2016

Botter, G., Bertuzzo, E., & Rinaldo, A. (2011). Catchment residence and travel time distributions: The master equation. Geophysical Research Letters, 38(11). http://doi.org/10.1029/2011GL047666

van der Velde, Y., Torfs, P. J. J. F., van der Zee, S. E. A. T. M., & Uijlenhoet, R. (2012). Quantifying catchment-scale mixing and its effect on timevarying travel time distributions. Water Resources Research, 48(6), W06536. http://doi.org/10.1029/2011WR011310

McMillan, H., Tetzlaff, D., Clark, M., & Soulsby, C. (2012). Do time-variable tracers aid the evaluation of hydrological model structure? A multimodel approach. Water Resources Research, 48(5). http://doi.org/10.1029/2011WR011688

Heidbüchel, I., Troch, P. a., Lyon, S. W., & Weiler, M. (2012). The master transit time distribution of variable flow systems. Water Resources Research, 48(6), W06520. http://doi.org/10.1029/2011WR011293

Davies, J., Beven, K., Rodhe, A., Nyberg, L., & Bishop, K. (2013). Integrated modeling of flow and residence times at the catchment scale with multiple interacting pathways, 49, 4738–4750. http://doi.org/10.1002/wrcr.20377

Bertuzzo, E., Thomet, M., Botter, G., & Rinaldo, A. (2013). Catchment-scale herbicides transport: Theory and application. Advances in Water Resources, 52, 232–242.

http://doi.org/10.1016/j.advwatres.2012.11.007

Hrachowitz, M., Savenije, H., Bogaard, T. a., Tetzlaff, D., & Soulsby, C. (2013). What can flux tracking teach us about water age distribution patterns and their temporal dynamics? Hydrology and Earth System Sciences, 17(2), 533–564. http://doi.org/10.5194/hess-17-533-2013

Harman, C., & Kim, M. (2014). An efficient tracer test for time-variable transit time distributions in periodic hydrodynamic systems. Geophysical Research Letters, 41(5), 1567–1575. http://doi.org/10.1002/2013GL058980

Soulsby, C., Birkel, C., & Tetzlaff, D. (2014). Assessing urbanization impacts on catchment transit times. Geophysical Research Letters, 41(2), 442–448. http://doi.org/10.1002/2013GL058716

van der Velde, Y., Heidbüchel, I., Lyon, S. W., Nyberg, L., Rodhe, A., Bishop, K., & Troch, P. a. (2014). Consequences of mixing assumptions for time-variable travel time distributions. Hydrological Processes. http://doi.org/10.1002/hyp.10372

Harman, C. J. (2015). Time-variable transit time distributions and transport: Theory and application to storage-dependent transport of chloride in a watershed. Water Resources Research, 51(1), 1–30. http://doi.org/10.1002/2014WR015707

Klaus, J., Chun, K. P., McGuire, K. J., & McDonnell, J. J. (2015). Temporal dynamics of catchment transit times from stable isotope data. Water Resources Research, 51(6), 4208–4223. http://doi.org/10.1002/2014WR016247

Benettin, P., Kirchner, J. W., Rinaldo, A., & Botter, G. (2015). Modeling chloride transport using travel time distributions at Plynlimon, Wales. Water Resources Research, 51(5), 3259–3276. http://doi.org/10.1002/2014WR016600

Hrachowitz, M., Fovet, O., Ruiz, L., & Savenije, H. H. G. (2015). Transit time distributions, legacy contamination and variability in biogeochemical $1/f\ \alpha$ scaling: how are hydrological response dynamics linked to water quality at the catchment scale? Hydrological Processes, 29(25), 5241–5256. http://doi.org/10.1002/hyp.10546

Birkel, C., Soulsby, C., & Tetzlaff, D. (2015). Conceptual modelling to assess how the interplay of hydrological connectivity, catchment storage and tracer dynamics controls nonstationary water age estimates. Hydrological Processes, 29(13), 2956–2969. http://doi.org/10.1002/hyp.10414

Soulsby, C., Birkel, C., Geris, J., Dick, J., Tunaley, C., & Tetzlaff, D. (2015). Stream water age distributions controlled by storage dynamics and nonlinear hydrologic connectivity: Modeling with high-resolution isotope data. Water Resources Research, 51(9), 7759–7776. http://doi.org/10.1002/2015WR017888

Benettin, P., Bailey, S. W., Campbell, J. L., Green, M. B., Rinaldo, A., Likens, G. E., McGuire, K. J., & Botter, G. (2015). Linking water age and solute dynamics in streamflow at the Hubbard Brook Experimental Forest, NH, USA. Water Resources Research, 51(11), 9256–9272. http://doi.org/10.1002/2015WR017552

REFERENCES WE USED:

Birkel, C., Soulsby, C. and Tetzlaff, D.: Conceptual modelling to assess how the interplay of hydrological connectivity, catchment storage and tracer dynamics controls nonstationary water age estimates, Hydrol. Process., 29(13), 2956–2969, doi:10.1002/hyp.10414, 2015.

Brooks, J. R., Wigington, P. J., Phillips, D. L., Comeleo, R. and Coulombe, R.: Willamette River Basin surface water isoscape ($\delta$ 18 O and $\delta$ 2 H): temporal changes of source water within the river, Ecosphere, 3(5), art39, doi:10.1890/ES11-00338.1, 2012.

[revised manuscript text omitted]

---

## Author Comment (AC2) · 29 Mar 2016

**Referee 2:**

The manuscript "Insights on the water mean transit time in a high-elevation tropical ecosystem" by Mosquera et al. under review in Hydrol. Earth Syst. Sci. Discuss., doi:10.5194/hess-2015-546, 2016 presents an attempt to investigate MTTs of a nested paramo catchment system in Ecuador with the purpose to tease out dominant controls on water transit time. The authors were able to identify relatively short transit times (

the paper generally well-written and structured. Having said that, the paper struggles in parts to clearly convey the main points in line with the objectives of the study and could be shortened. I am missing a discussion around arguments that the MTT is not a meaningful catchment descriptor and the recent tendency towards the recognition of the time-variant nature of transit times. I do think that there are merits in using the MTT to characterize catchment systems particularly considering the constraints and limitations working in tropical environments; it should, however, be more clearly argued. Furthermore, there are some model decisions that should be more clearly explained, which also likely leads to additional analysis strengthening the paper and its line of arguments. Nevertheless, I think this is nothing that cannot be fixed with a careful revision to improve clarity and focus of the paper and I therefore support publication of this paper with some revisions.

Reply: We appreciate Christian Birkel's (R2) revision and his constructive suggestions to improve the scientific quality of the manuscript and look forward to publishing this work in HESS hoping to improve the understanding of hydrologic processes in tropical ecosystems. We agree that the paper could be shorter to focus its content in the objectives. We also acknowledge that some details of the modeling procedure deserve clarification along with a discussion about recent theoretical frameworks that explicitly incorporate time-variant transit times (please see response to R1). Our responses to R2 comments are outlined below.

Specific comments:

Abstract:

Line 21: I'm not sure if the paper is about streamwater MTT as you excluded high-flow events from the analysis. Reply: We agree, based on the discussion below, it is clear that MTT estimations correspond to baseflow MTTs. This will be specified throughout the manuscript.

Key words:
Line 15: I suggest to simplify and reduce the key words to attract more online search results, e.g.: Ecohydrology, MTT, runoff generation, Andean paramo, Histosols, Ecuador.

Reply: We agree with the suggestion and will simplify and reduce the key words to the following: Ecohydrology, MTT, runoff generation, wet Andean páramo, high altitude tropical wetlands, Histosol, Ecuador

Introduction:

Page 3, Line 2: This is true for Latin America, but there are a few more studies in the tropics. You could even refer to Muñoz-Villers and McDonnell (2012) in this context.

Reply: We agree and will include these additional references: (Farrick and Branfireun, 2015; Muñoz-Villers et al., 2015) of studies conducted in the tropics.

Page 3, Line 17: I will come back to this point, but I think it's very likely that there's also a considerable near-surface runoff component as seen in other environments (you refer to Scotland and Sweden below) with organic rich wetland soils that remain saturated for much of the year. I, however, don't know the paper in review you cite here.

Reply: We see that this point can cause confusion. However, by "shallow subsurface flow" we refer to water moving in the first 30-40 cm within the soil matrix (i.e., "shallow organic horizon of the páramo soils located near the streams"). This will be clarified in the text. This is supported by water isotopic concentrations observed in the shallow organic horizon of the Histosol soils and stream waters in the study area. These observations and further discussion are presented in a manuscript currently under review in Hydrological Processes. We will provide the editor of this paper with a copy the referred manuscript to pass it along to the reviewers.

Page 3, Line 29: This isn't entirely true, I'm afraid, because the dominating runoff generation process based on various tracer studies is a rapid near-surface flow. The subsurface component is a deeper and slower groundwater flux. Therefore, the wet-land contribution can be quantified very well in form of near-surface saturation overland

HESSD
flow.

Reply: We partially agree with this comment. The hydrologic functioning of this particular system takes place in the first 30-40 cm of the soils, here referred to as "shallow subsurface flow", and saturation excess overland flow rarely occurs in the Zhurucay basin (i.e., high flows occur less than 3% of the time at the study site, see figure 3 in Mosquera et al., 2015). Although it is true that "rapid near-surface flow" has been observed in other environments, it mostly refers to "near-surface saturation overland flow", and therefore, it does not apply to our system. We will clarify this point by changing the term "subsurface" by "shallow subsurface" in the revised version of the manuscript.

Page 4, Line 9: I think Broxton et al. (2009) worked in Arizona, USA. You could also specify the control you are referring to as in this case it was "aspect".

Reply: We appreciate the suggestion. We will incorporate this reference in the manuscript and specify the controls on MTT.

Study site:

Page 5, Line 8: This is an awkward sentence, please revise.

Reply: We agree and will revise and restructure the sentence.

Line 10: seasonality, primarily.

Reply: These suggestions will be incorporated.

Line 23: please, spell out INV.

Reply: INV is the name of the mining company (http://www.invmetals.com/about/history/). We will remark this in the manuscript.

Page 6, Line 4-5: Please, revise this sentence.

Reply: The sentences referring to soil distribution in the catchment will be restructured.

Line 27: Please, indicate model and make of the equipment.
Reply: Model and make of the instrument (Schlumberger DI500) will be indicated. Methods:

Page 9, Line 26: ... is based...

Reply: We agree with this suggestion and will correct accordingly.

Line 27: I'm not sure I follow the second point.

Reply: This refers to input (recharge) function of the precipitation tracer composition to take into account recharge (i.e., volumetrically weighted isotopic composition) (McGuire and McDonnell, 2006). This will be clarified.

Page 10, Line 6: In this case, I suggest to consistently refer to a baseflow MTT and not streamwater MTT.

Reply: We agree. The following will be added: "As a result, estimations correspond to baseflow MTTs, hereafter simply referred to as MTT."

Page 11, Line 2-5: I fully agree that you seek to identify the best-performing and most parsimonious model. However, you don't really compare the models using a criterion for model selection (e.g., AIC, BIC or adjusted R2) that penalizes the number of parameters in combination with a goodness-of-fit measure. The MI criterion looks at how identifiable one parameter is, but not at the combined effect of more than one parameter used to calibrate the model.

Reply: We appreciate this comment. We applied the AIC metric for model selection to our results and will include the results from this analysis in the manuscript. This analysis indicates that the exponential model (EM) is indeed the best.

Page 11: How were models generated? Using a uniformly sampled Monte Carlo procedure?

Reply: This is correct. The fitting procedure included two steps for each model. 1)
Initially 10,000 sets of parameter values were evaluated considering a wide range of parameter values sampled according to a uniform Monte Carlo procedure. The parameter ranges were wide. For instance the parameter range of the MTT of the EM model varied between 0 and 130 biweeks (5 yrs). 2) After the initial 10,000 runs, the range of the set of parameters that displayed relatively well identified were narrowed and the model was run again until 1,000 behavioral parameter sets were obtained (i.e., sets of parameters that yielded solutions corresponding to at least 95% of the highest KGE).

Line 16: mainly?

Reply: We appreciate the suggestion "Majorly" will be changed to "mainly".

Results:

Page 12, Line 12: Runoff coefficients show...

Reply: This sentence will be updated accordingly.

Page 13, Line 4-26: I'm not convinced by some of the statements present in this paragraph. For example, the best-fit gamma model compared to the best-fit exponential model does show a quite significant increase in performance (from 0.63 to 0.75) that can justify the use of one additional fitting parameter. On the other hand, a third fitting parameter resulted in an increased performance of only 0.01. The poorest model seems to be the DM with a best-fit of KGE=0.5. Based on this, one could qualitatively reject the DM and TPLR models as suitable models compared to the EM and GM. However, the decision between the EM and GM models should be informed by a model selection criterion such as the AIC (see comment above) that evaluates the combined effect of the parameters on model performance.

Reply: We completely agree and appreciate the suggestion. As indicated above we have now conducted an AIC evaluation that confirms our model selection. The EM model is indeed the one the yields the lowest AIC score.

Page 14, Line 12: Please, revise this sentence.
Reply: Thank for the suggestion, the sentence will be revised and corrected.

Page 14: I think that large parts here could be moved into the discussion or simply be deleted as later sections pick up on these issues. This would allow to shorten the m/s focussing on presenting the key results and later discussion in the light of the wider literature.

Reply: We agree with this suggestion. We will trim down irrelevant text in this section. In particular, we will consider removing or attaching the section regarding the pdf and cdf TTD curves as supplementary material.

Page 15, Line 26: just use MTT

Reply: This will be corrected accordingly.

Page 16, Line 5-12: Please, separate this very long sentence into smaller parts.

Reply: We appreciate the suggestion. This sentence will be split into three.

Discussion:

Page 17, Line 26: more depleted?

Reply: You are correct, this will be updated.

Page 18, Line 1: I think it would be better to indicate that baseflow MTT was analysed.

Reply: We agree and will change this accordingly.

Line 3: identifiability?

Reply: We agree.

Line 20: Was TMCF previously defined?

Reply: Thank you catching this. A definition will be included.

Line 30: remain.

HESSD
Reply: We agree.

Page 19, Line 10: You somehow have to convince me that this actually is subsurface stormflow. I haven't seen the in review paper you mention in this context and all the evidence you show tells me that the dominating runoff generation mechanism is near-surface saturation overland flow due to little mixing with deeper soil horizons, short MTTs, etc.

Reply: Based on the characterization of the weekly isotopic composition of stream and soil waters conducted over a two-years period, it is evident that the isotopic composition of the shallow organic horizon of the Histosol soils consistently matches that of the streams, and that precipitation has essentially no influence in the streamflow isotopic composition (Mosquera et al., 2016). As such, even "subsurface stormflow" appears to inappropriately describe the system's functioning, as water is preferentially delivered from the shallow 30-40 cm of the organic horizon of the Histosols to the streams regardless of the precipitation dynamics. Moreover, saturation excess overland flow rarely occurs at the study site (Mosquera et al., 2015). Therefore, we consider that "shallow subsurface flow" is indeed the appropriate term to define the delivery of water from the Histosols situated at the bottom of the slopes to the streams. We will include a clearer explanation of this.

Line 20: You previously said up to 2.2 years in this context.

Reply: We referred to two different studies conducted in central Mexico. Muñoz-Villers and McDonnell (2012) reported MTTs of three years and recently Muñoz-Villers et al. (2015) reported MTTs ranging between 1.2-2.2 years. Therefore, this statement in the manuscript is correct.

Page 20, Line 6: I think Hrachowitz et al. (2009) argued with saturation overland flow.

Reply: Hrachowitz et al. (2009) actually reported that at the Lord Arch catchment runoff generation shows a flashy catchment response "dominated by runoff processes in the
upper soil horizons." That is, in the 40 cm depth peaty soils, overlaying the mineral horizons,

Line 11: solutes.

Reply: Agree.

Page 22, Line 1: explain.

Reply: Agree.

Line 8: Please, revise this sentence.

Reply: We agree. This sentence will be reworded.

Line 14: Isn't this simply the slope?

Reply: Yes, you are correct. We will change it accordingly.

Line 24: I find the "regulation capacity" is coming a bit out of nowhere. What exactly do you mean by this? Is it in the sense of resilience or simply that the turn-over is quick and what goes in comes out with little delay?

Reply: We acknowledged how this can lead to confusion. We will clarify this in the text. Basically the páramo is an ecosystem recognized for its high discharge regulation capacity (i.e., páramo generates runoff year-round regardless of variability in precipitation inputs to the system). This characteristic is essential to the sustainability of human activities of downstream populations. However, little is known about the factors driving this regulation capacity. The results from this study provide information that improves our understanding of catchment functioning by identifying some of these drivers. That is, the interplay between soil storage and topography. We will explain the regulation capacity notion of the ecosystem in the introduction section of the manuscript.

Page 23, Line 2: It's the first time that you mention that SOF wasn't previously observed in the study catchment. This information needs to come earlier. I also think this whole
**HESSD**
paragraph can be shortened towards the key messages presented at the very end.

Reply: We agree. We will add information regarding SOF earlier in the manuscript. In addition we will trim this paragraph to reduce its length.

Line 29: Please, revise this sentence.

Reply: We agree, we will reword the sentence.

Tables:

Shouldn't the current Table 2 come before you present the models (Table 1)?

Reply: We appreciate this suggestion and completely agree.

Current Table 1: I'm a bit confused about some decisions concerning the choice of initial parameter intervals. Why was the upper limit of tau set at 200 biweeks? This makes 2800days and over 7 yrs of TT, something stable isotopes aren't able to detect anyways (Stewart et al., 2010). Further, why was the lower limit of beta (GM) set to 0.5? In the case of low TT this could be well below 0.5 and on a global scale the average resulted to be at around 0.5 (Godsey et al., 2009). With the current lower limit in place you potentially miss suitable parameters that would also result in lower MTTs compared to current best-fit results; an argument you used to reject the GM. Also, it seems odd to me that you don't report the parameter interval for beta as this is the parameter you calibrate. The MTT (tau) is only the result of beta\*alpha.

Reply: We really appreciate this comment. We used the MTT parameter ranges suggested by Timbe et al., (2014). However, we recognize that these authors had a different objective in their study and that it is reasonable that we constrain our parameter values range for MTT up to 5 years (130 weeks) (McGuire and McDonnell, 2006). As such, we will run all models again for all catchments, and statistics and figures will be updated accordingly. Regarding the parameter in the GM, we believe R2 refers to the alpha parameter. The alpha parameter lower limit was originally set up at 0.01, and the 0.5 value was just mistakenly reported in the table. We apologize for the confusion this HESSD
caused. We will correct the lower limit value in the table and also report the parameter range considered for beta.

Table 5: Similar issue here with the GM. I suggest to report the parameters alpha and beta.

Reply: Same as above. Beta parameter will be reported in the figure.

Table 7: R2-values of 0.62 did not result significant? However, there's a relationship with flow characteristics particularly for the extremes and the runoff coefficient does seem to explain some of the spatial variability among catchments.

Reply: We appreciate this comment. The mentioned relations are not statistically significant at a 95% confidence level. Results are as follows: Runoff coefficient: R2 0.62 p-value: 0.39 Q99: R2 -0.42 p-value 0.24 Q10: R2 -0.61 p-value 0.12 Q5: R2 -0.62 p-value 0.11 However, we agree that there is a relation between baseflow MTTs and low flows that deserves to be considered. Thus, p-values will be included in table 7 and relation with low flows will be discussed accordingly.

Figures:

Figure 2: What's the purpose of the streamflow inlet box? Could you not just show a log-scale to emphasize the low flow periods? Those event samples do show quite a bit of response to rainfall. What's the effect of pooling these out? Quite a bit shorter MTTs? Please, consider adjusting the different EC sampling period for comparison purposes.

Reply: 1) Streamflow inlet box: The purpose of the streamflow inlet box is to emphasize the response of low flows to rainfall inputs during the less humid periods. The box indicates flashy response even during these periods. We therefore still believe that the non-log-scale representation of the hydrograph in combination with the inlet box provides the best impression of the observed dynamics. 2) Event samples: The model runs reported were originally conducted once these referred event samples were Interactive comment

pooled out from the streamflow isotopic composition time series. 3) EC sampling period: Sampling period for EC will be adjusted to hydrometric and isotopic data sampling period.

Figure 5: Please, clarify if sampling was started below alpha = 0.5 (GM) contrary to the information from Table 1. Again, I suggest to present the parameters alpha and beta.

Reply: We apologize for the confusion. The lower limit of the alpha parameter was originally set up at a value of 0.01. This was updated in Table 2. We now present both alpha and beta parameters as suggested, together with the MTT.

Figure 6: Is EPM missing in the right panel?

Reply: No, it is not. It just plots behind the EM curves in both panels. A note will be added to the caption to explain this.

Figure 8: If the MTT is normalized shouldn't it be unitless?

Reply: You are correct. We will change accordingly in figures 8 and 9.

References I used:

Godsey, S. E., Aas, W., Clair, T. A., de Wit, H. A., Fernandez, I. J., Kahl, J. S., Malcolm, I. A., Neal, C., Neal, M., Nelson, S. J., Norton, S. A., Palucis, M. C., Skjelkvåle, B. L., Soulsby, C., Tetzlaff, D. and Kirchner, J. W. (2010), Generality of fractal 1/f scaling in catchment tracer time series, and its implications for catchment travel time distributions. Hydrol. Process., 24: 1660 1671. doi: 10.1002/hyp.7677.

Stewart MK, Morgenstern U, McDonnell JJ. 2010. Truncation of stream residence time: How the use of stable isotopes has skewed our concept of streamwater age and origin. Hydrological Processes 24: 1646–1659.

**REFERENCES WE USED:**

Farrick, K. K. and Branfireun, B. A.: Flowpaths, source water contributions and water
residence times in a Mexican tropical dry forest catchment, J. Hydrol., 529, 854–865, doi:10.1016/j.jhydrol.2015.08.059, 2015.

Hrachowitz, M., Soulsby, C., Tetzlaff, D., Dawson, J. J. C., Dunn, S. M. and Malcolm, I. A.: Using long-term data sets to understand transit times in contrasting headwater catchments, J. Hydrol., 367(3-4), 237–248, doi:10.1016/j.jhydrol.2009.01.001, 2009.

McGuire, K. J. and McDonnell, J. J.: A review and evaluation of catchment transit time modeling, J. Hydrol., 330(3-4), 543–563, doi:10.1016/j.jhydrol.2006.04.020, 2006.

Mosquera, G. M., Lazo, P. X., Célleri, R., Wilcox, B. P. and Crespo, P.: Runoff from tropical alpine grasslands increases with areal extent of wetlands, CATENA, 125, 120–128, doi:10.1016/j.catena.2014.10.010, 2015.

Mosquera, G. M., Célleri, R., Lazo, P. X., Vaché, K. B. and Crespo, P. J.: Combined Use of Isotopic and Hydrometric Data to Conceptualize Ecohydrological Processes in a High-Elevation Tropical Ecosystem, Hydrol. Process., 2016.

Muñoz-Villers, L. E. and McDonnell, J. J.: Runoff generation in a steep, tropical montane cloud forest catchment on permeable volcanic substrate, Water Resour. Res., 48(9), n/a–n/a, doi:10.1029/2011WR011316, 2012.

Muñoz-Villers, L. E., Geissert, D. R., Holwerda, F. and McDonnell, J. J.: Factors influencing stream water transit times in tropical montane watersheds, Hydrol. Earth Syst. Sci. Discuss., 12(10), 10975–11011, doi:10.5194/hessd-12-10975-2015, 2015.

Timbe, E., Windhorst, D., Crespo, P., Frede, H.-G., Feyen, J. and Breuer, L.: Understanding uncertainties when inferring mean transit times of water trough tracer-based lumped-parameter models in Andean tropical montane cloud forest catchments, Hydrol. Earth Syst. Sci., 18(4), 1503–1523, doi:10.5194/hess-18-1503-2014, 2014.

**HESSD**

---

## Author Response (AR1)

**Point-by-point response to the Referee 1's review**

Reply: We really appreciate the review provided by referee 1 (R1) and are glad that our work gives rise to an interesting discussion of catchment heterogeneity and non-stationarity in the context of robust MTT estimations. R1 comments focus on four main aspects: 1) site conditions, 2) aggregation bias, 3) model evaluation and performance, and 4) selected methodology. Hence, we provide responses to each of them separately below.

R1: The paper attempts to explore transit time distributions (TTD) in a high-elevation tropical ecosystem by using a detailed hydrologic and isotopic record from eight nested catchments located in southern Ecuador.

Reply: First, we want to emphasize that our system is characterized by unique hydrometeorological and landscape characteristics in comparison to other systems: i) mean annual precipitation, runoff, and evapotranspiration are similar across the entire catchment (1284±18 mm yr$^{-1}$, 788±54 mm yr$^{-1}$, 496±61 mm yr$^{-1}$, respectively) (Mosquera et al., 2015), ii) precipitation is evenly distributed year-round with very low degree of seasonality (Padrón et al., 2015), iii) isotopic fractionation by evaporative effects is virtually negligible (Mosquera et al., 2016) as a result of the year-round high relative humidity (~90%) (Córdova et al., 2015), iv) the soils are shallow and poorly developed across the entire catchment (~1m deep), and v) the geology is relatively young and homogeneous (Coltorti and Ollier, 2000). We believe our system presents a high degree of homogeneity across the entire basin as a result of the mentioned landscape configuration and local hydrometeorological conditions as is noted in Page.9, Lines. 16-19 (P. 9, L. 16-19).

R1: Although the data are extremely interesting and unique in quality and location, the transit time analysis is performed through a method (the lumped convolution approach) which is likely to include an aggregation bias, especially for systems with a high degree of heterogeneity and non-stationarity (see the recent papers by Kirchner, [2016a,b]).

Reply: We appreciate the comment and have significantly expanded the text to more fully recognize recent advances in MTT estimation, and more clearly place the paper within them. Regarding our approach, we will first focus on the issue of heterogeneity. It is certainly true that heterogeneity is "a fundamental problem" in the investigation of catchment behavior, because the scale of investigation influences the type of hydrological processes that can be identified. This issue is very well captured in Figure 4 of Kirchner (2016a) which exemplifies how MTT estimations can be affected as a result of aggregation across scales in heterogeneous catchments. We recognized this issue in our study design by considering a nested monitoring configuration. This configuration allowed us to investigate the variability of hydrological processes across scales and to characterize the system's degree of heterogeneity. As mentioned above, the characteristics of the Zhurucay basin, provide quasi-homogeneous conditions or low degree of heterogeneity, which most likely significantly reduces the issue of aggregation bias in MTT estimations. The landscape homogeneity in our system is evident considering that the same TTD represent the subsurface hydrologic system's behavior at all catchments, with a relatively small range of variation of the estimated MTTs (0.51±0.17 yr). Thus, we consider that given the homogeneous catchment characteristics and small seasonality of local environmental conditions in our system, applying the LCA to our water stable isotopic dataset is a first step with which to build-up improved catchment functioning understanding.

Regarding the nonstationarity issue (Kirchner, 2016b), we first want to clarify that using a certain methodology does not imply that other ones are ignored. The LCA is an approach that assumes steady-state conditions in the system, which we explicitly acknowledge in our paper (P. 9, L. 18-19 and P. 11, L. 13-15). However, as highlighted in the manuscript title (i.e., "Insights"), our study aims to set a baseline for the application of modeling techniques using stable water isotopes in tropical ecosystems above the tree line. We make the case that this is an important feature of the earth system, in which there is generally only very scarce hydrologic information. In fact, to our knowledge, this is the first contribution regarding the modeling of MTT in páramo ecosystems.

Although recent advances in hydrologic research, such as the ones listed by R1, have provided theoretical evidence of the importance of recognizing the unsteady nature of hydrologic processes highlighting the possible shortcomings of the LCA, our paper is not designed as a methodological contribution regarding MTT modeling theory. Instead, this paper is about understanding catchment functioning in a high-elevation tropical region. Indeed, one of our future goals is to apply modeling techniques that explicitly recognize non-stationarity and storage dynamics in the hydrological behavior of this tropical ecosystem (e.g.,Birkel et al., 2015; Harman, 2015; Hrachowitz et al., 2013). These analyses, however, are beyond the scope of this paper. That said, we also believe our data set is valuable because it provides a concrete example to test the applicability, limitations, and constraints of different MTT modeling methodologies. In fact, based on the homogeneous and uniform hydrometeorological characteristics of our site, in contrast to many of the temperate regions in which time-variant MTT modeling has been developed, we anticipate our dataset and the results from this study to allow for benchmark testing of MTT methodologies in regions with low climate seasonality. At the same time, we acknowledge the growing recognition of the time-variant nature of transit times and highlight the value of such modeling methodologies and recent findings related to them in P. 4, L.20 to P5. L7.

It is relevant to note a recent application of conceptual modeling for the investigation of non-stationary conditions in a wet Scottish upland catchment where runoff generation processes mainly occur in the riparian Histosol soils with high storage capacity (Birkel et al., 2015), as in our study site. These authors detected non-stationary characteristics in water age distributions only during extreme weather conditions (extensive dry or wet periods) and attributed this behavior to the large mixing capacity of the Histosol soils, "which acts as an isostat moderating isotope variability and limiting the time variance of water age". The latter has been clearly observed in the Zhurucay basin, where the isotopic composition of the organic horizon of the Histosol soils remains virtually constant and matches the isotopic composition in the streams year-round (Mosquera et al., 2016). This, in combination with the nearly uniform climate characteristics in our site, supports the utility of steady-state approaches in our system. We have included a discussion of these findings in relation to the ones at our site in P. 21, L. 4-13.

R1: In simple terms, even if the transfer function approach allows a fair simulation of the measured isotopic signal, the system mean transit time is not necessarily realistic, due to the structural uncertainties in the quantification of the older water components. This emerges in Figures 4 and 6, where different TTD (with different MTT) result in similar model performances.

Reply: With regards to the uncertainties related to the quantification of old water components, we consider this is not a significant issue in our system. Evidence of very low (insignificant) "old" water contributions (i.e., deep groundwater contributions to discharge) in the Zhurucay basin has been found by Crespo et al. (2011) and Mosquera et al. (2016) and by Buytaert and Beven (2011) in a nearby páramo catchment in South Ecuador. It appears that this results from the combination of: 1) the relatively young and homogeneous geology, 2) the high storage capacity of the porous organic horizon of the páramo soils, and 3) their low level of development (soils are generally less than 1 m deep). Results from our study support this interpretation. This is evidenced by the fact that the two TTDs functions (Gamma and two parallel linear reservoir) that incorporate an "old" water component yield parameter values that are not well constrained (see P. 19, L. 6-11). Instead the exponential model provided a robust representation with a clearly defined parameter. In this sense, it must be highlighted that our procedure to identify the TTD that best describes our system hydrologic behavior did not only take into account the goodness-of-fit of the objective function but also the level of identification of the function parameters and a process-based interpretation of the results (see below).

Regarding the similar performance of the model using different TTDs, Timbe et al. (2014) conducted a detailed analysis of the uncertainties related to the use of different TTDs in MTT modeling using the LCA. They also found that several TTDs provide high goodness-of-fit between predictions and observations, but poor parameter identifiability for some TTDs calibrated parameters, as has also been observed by other researchers (e.g., Hrachowitz et al., 2009). As such they recommend that for achieving meaningful MTT estimates from the LCA, it is at least needed to: 1) used several TTDs, 2) evaluate predictions uncertainty, and 3) assess parameter identifiability for each TTD function. Following these recommendations, we conducted an assessment of the performance of different TTDs, considering not only the best fit but also the uncertainty of the predictions, the parameter identifiability, and a process-based interpretation in light of the detailed hydrometric, isotopic, and biophysical landscape information which has been collected at our study site over the last five years (Córdova et al., 2015; Mosquera et al., 2015, 2016; Padrón et al., 2015; Quichimbo et al., 2012) to select the model that best describes the hydrologic functioning of the system. In addition, we have included an analysis using a metric for model selection (Akaike information criterion, AIC). This analysis has confirmed the EM as the one that best describes the hydrologic functioning in our system (see P. 14, L. 21-24).

R1: Moreover, the paper ignores the recent advances in hydrologic transport and TTD (see the list of suggested literature), which are now widespread within the hydrologic community and have clarified the concept of TTD in the light of non-stationarity. The manuscript is clear, well written and easy to follow, but the methods pose some serious concern on the paper's conclusions.

Reply: We appreciate this comment and have included a clear discussion about the recent time variant advances in hydrologic modeling in P. 4, L.20 to P. 5, L. 21. However, given the high degree of homogeneity of our system we believe that assuming steady-state conditions is justified. This conditions allows us to use LCA to develop an improved understanding of catchment function using hydrometric-tracer based hydrologic modeling. There is information in the LCA approach, information which we explore and highlight in this paper. Limiting transit time modeling efforts only to recent methodologies suggests that these baseline MTT

estimates, developed with a well-established approach (with well-established limitations) are of no value. We disagree wholeheartedly. MTT modeling under non-stationary conditions is an idea that is currently under development, and as result, there is yet no unified methodology that can be globally applied. Indeed there are very few applications of such methodology, most of which have yield results with high degree of uncertainty (Harman, 2015; Klaus et al., 2015; McMillan et al., 2012) or it is not even estimated (Davies et al., 2013; Heidbüchel et al., 2012; van der Velde et al., 2015), mainly as a result of expensive computation costs or high uncertainties related to the spatial variability of the input hydrometric and tracer field measurements. It is clear however, that given the mathematical limitations (Duvert et al., 2016; Seeger and Weiler, 2014), high-temporal resolution of tracer data required (Harman, 2015; Heidbüchel et al., 2012), and general unavailability of long-term tracer records (Hrachowitz et al., 2010; Klaus et al., 2015) also required for hydrological modeling under non-stationary conditions, the LCA is still a useful metric of storage and catchment functioning not only in understudied regions such as the tropics (e.g., Farrick and Branfireun, 2015; Muñoz-Villers et al., 2015; Timbe et al., 2014) but also elsewhere (e.g., Duvert et al., 2016; Hale and McDonnell, 2016; Hale et al., 2016; Hu et al., 2015; Seeger and Weiler, 2014). In this sense, we agree with Christian Birkel (referee 2, R2) comment: "there are merits in using the MTT to characterize catchment systems particularly considering the constraints and limitations working in tropical environments" and are convinced that this "experimentally derived dataset for this tropical ecosystem is unique and interesting to the HESS readership and beyond", particularly taking into account the system´s particular characteristics. We believe this contribution will become a benchmark study over which to build-up further hydrological processes understanding not only in this remote understudied region, but also more generally, in regions with low climate seasonality and catchments with low degree of heterogeneity. Future efforts will built upon the monitoring infrastructure and datasets that continue to be collected in the Zhurucay River Ecohydrological Observatory, which will allow for continual improvement in hydrologic interpretation by eventually incorporating some alternative modeling techniques (e.g., Birkel et al., 2015; Harman, 2015; Hrachowitz et al., 2013).

DETAILED COMMENTS

Page 7, line 18: the authors say that kinetic fractionation by evaporation can be neglected, however looking at Figure 3 it seems that the majority of stream water samples plot below the LMWL. How can this behavior be explained?

Reply: You are correct. Fractionation by evaporation is not negligible. A detailed analysis of the deuterium-oxygen-18 relation in rainfall and stream waters in the Zhurucay basin has shown that kinetic fractionation by evaporation is yet very low (Mosquera et al., 2016). However, Figure 3 has been removed as the information it conveyed is not relevant for the discussion of the paper and a thorough discussion of such information can be found in the manuscript of Mosquera et al. (2016), currently under review and accepted with minor revisions in Hydrological Processes.

P. 9, l. 3: the variable tau in Eq. (1) and (2) is not the mean transit time. It is just the dummy variable in the integral, which spans the transit time domain [0, +inf].

Reply: Thank you catching this typo. We have make a correction by removing this from the manuscript.

P. 10, l. 22-26: I did not get why the model is run twice to get the behavioral set of parameters.

Reply: We first run the model 10,000 times considering a wide range of parameter values. This provided and idea about the range of acceptable values. Based on the latter, we narrowed the parameter space and run the model again until 1,000 solutions or more, corresponding to at least 95% of the KGE objective function (i.e., at least 1,000 behavioral solutions), were obtained. These 1000 solutions allowed strong identification of the 90% confidence interval using the GLUE methodology. This procedure is been clarified in P. 12, L. 4-9.

P. 10, l. 28: the MI index seems to be very arbitrary depending on the choice of the prior parameter distribution. Segura et al., [2012] provide a partial explanation for their choice of the prior, which is here missing.

Reply: We appreciate this observation. Our initial choice of parameters ranges was selected based on the work of Timbe et al. (2014) in their analysis of uncertainties related to the use of the LCA for MTT estimations. This is been clarified this in P. 12, L. 16-18.

P. 13, l. 27: the terminology "MTT probability density function" seems to refer to the pdf of MTT obtained from the posterior parameter distribution.

Reply: You are correct. The pdf and cdf we referred to in the text correspond to the distributions described based on the fitted parameter distributions. This is been clarified in the text in P. 16, L. 19.

P. 14, l. 1-13: this is to me a clear example of the indetermination of the MTT. Different parameterizations of the TTD are able to provide good, similar simulations of the isotopic signal, but result in rather different MTT. While it is reasonable to choose a model because its parameters are more constrained in the simulation of a specific target, this does not allow to extrapolate that its MTT is the "right" one.

Reply: This issue is been already discussed above in the general comment and we just want to emphasize that we carefully considered the uncertainty of the predictions and the parameter identifiability, in addition to the results of the simulation of the isotopic signal against the objective function and have further included the results from the AIC model selection metric that support that the EM is the one that best describes our system in P. 14, L. 21-24 and Table 5, as suggested by R2. Additionally, we conducted a process-based interpretation of these results in light of the detailed hydrometric, isotopic, and biophysical landscape information which has been collected at the Zhurucay basin over the last five years (Córdova et al., 2015; Mosquera et al., 2015, 2016; Padrón et al., 2015; Quichimbo et al., 2012) for selecting the TTD that best suites the hydrologic conditions of the system.

P. 15, l. 21: what is meant by "completely" recovered? Is there a threshold (e.g. 99%) on the recovered mass?

Reply: This means that if the tracer would have been injected as a single pulse, how much it would take to completely leave the system. In effect, certain proportion of the total injection will be recovered at a certain time after the injection, For example, Figure 6 depicts that for the EM 80% of the tracer is recovered at around 20 biweeks. Analogous analysis have been reported by Hrachowitz et al. (2009) and McGuire et al. (2005).

SUGGESTED LITERATURE

Kirchner, J. W. (2016a). Aggregation in environmental systems - Part 1: Seasonal tracer cycles quantify young water fractions, but not mean transit times, in spatially heterogeneous catchments. Hydrology and Earth System Sciences, 20(1), 279–297. http://doi.org/10.5194/hess-20-279-2016

Kirchner, J. W. (2016b). Aggregation in environmental systems - Part 2: Catchment mean transit times and young water fractions under hydrologic nonstationarity. Hydrology and Earth System Sciences, 20(1), 299–328. http://doi.org/10.5194/hess-20-299-2016

Botter, G., Bertuzzo, E., & Rinaldo, A. (2011). Catchment residence and travel time distributions: The master equation. Geophysical Research Letters, 38(11). http://doi.org/10.1029/2011GL047666

van der Velde, Y., Torfs, P. J. J. F., van der Zee, S. E. A. T. M., & Uijlenhoet, R. (2012). Quantifying catchment-scale mixing and its effect on timevarying travel time distributions. Water Resources Research, 48(6), W06536. http://doi.org/10.1029/2011WR011310

McMillan, H., Tetzlaff, D., Clark, M., & Soulsby, C. (2012). Do time-variable tracers aid the evaluation of hydrological model structure? A multimodel approach. Water Resources Research, 48(5). http://doi.org/10.1029/2011WR011688

Heidbüchel, I., Troch, P. a., Lyon, S. W., & Weiler, M. (2012). The master transit time distribution of variable flow systems. Water Resources Research, 48(6), W06520. http://doi.org/10.1029/2011WR011293

Davies, J., Beven, K., Rodhe, A., Nyberg, L., & Bishop, K. (2013). Integrated modeling of flow and residence times at the catchment scale with multiple interacting pathways, 49, 4738–4750. http://doi.org/10.1002/wrcr.20377

Bertuzzo, E., Thomet, M., Botter, G., & Rinaldo, A. (2013). Catchment-scale herbicides transport: Theory and application. Advances in Water Resources, 52, 232–242. http://doi.org/10.1016/j.advwatres.2012.11.007

Hrachowitz, M., Savenije, H., Bogaard, T. a., Tetzlaff, D., & Soulsby, C. (2013). What can flux tracking teach us about water age distribution patterns and their temporal dynamics? Hydrology and Earth System Sciences, 17(2), 533–564. http://doi.org/10.5194/hess-17-533-2013

Harman, C., & Kim, M. (2014). An efficient tracer test for time-variable transit time distributions in periodic hydrodynamic systems. Geophysical Research Letters, 41(5), 1567–1575. http://doi.org/10.1002/2013GL058980

Soulsby, C., Birkel, C., & Tetzlaff, D. (2014). Assessing urbanization impacts on catchment transit times. Geophysical Research Letters, 41(2), 442–448. http://doi.org/10.1002/2013GL058716

van der Velde, Y., Heidbüchel, I., Lyon, S. W., Nyberg, L., Rodhe, A., Bishop, K., & Troch, P. a. (2014). Consequences of mixing assumptions for time-variable travel time distributions. Hydrological Processes. http://doi.org/10.1002/hyp.10372

Harman, C. J. (2015). Time-variable transit time distributions and transport: Theory and application to storage-dependent transport of chloride in a watershed. Water Resources Research, 51(1), 1–30. http://doi.org/10.1002/2014WR015707

Klaus, J., Chun, K. P., McGuire, K. J., & McDonnell, J. J. (2015). Temporal dynamics of catchment transit times from stable isotope data. Water Resources Research, 51(6), 4208–4223. http://doi.org/10.1002/2014WR016247

Benettin, P., Kirchner, J. W., Rinaldo, A., & Botter, G. (2015). Modeling chloride transport using travel time distributions at Plynlimon, Wales. Water Resources Research, 51(5), 3259–3276. http://doi.org/10.1002/2014WR016600

Hrachowitz, M., Fovet, O., Ruiz, L., & Savenije, H. H. G. (2015). Transit time distributions, legacy contamination and variability in biogeochemical 1/f α scaling: how are hydrological response dynamics linked to water quality at the catchment scale? Hydrological Processes, 29(25), 5241–5256. http://doi.org/10.1002/hyp.10546

Birkel, C., Soulsby, C., & Tetzlaff, D. (2015). Conceptual modelling to assess how the interplay of hydrological connectivity, catchment storage and tracer dynamics controls nonstationary water age estimates. Hydrological Processes, 29(13), 2956–2969. http://doi.org/10.1002/hyp.10414

Soulsby, C., Birkel, C., Geris, J., Dick, J., Tunaley, C., & Tetzlaff, D. (2015). Stream water age distributions controlled by storage dynamics and nonlinear hydrologic connectivity: Modeling with high-resolution isotope data. Water Resources Research, 51(9), 7759–7776. http://doi.org/10.1002/2015WR017888

Benettin, P., Bailey, S. W., Campbell, J. L., Green, M. B., Rinaldo, A., Likens, G. E., McGuire, K. J., & Botter, G. (2015). Linking water age and solute dynamics in stream-flow at the Hubbard Brook Experimental Forest, NH, USA. Water Resources Research, 51(11), 9256–9272. http://doi.org/10.1002/2015WR017552

REFERENCES WE USED:

[revised manuscript text omitted]

**Point-by-point response to Christian Birkel (Referee 12)'s review**

The manuscript "Insights on the water mean transit time in a high-elevation tropical ecosystem" by Mosquera et al. under review in Hydrol. Earth Syst. Sci. Discuss., doi:10.5194/hess-2015-546, 2016 presents an attempt to investigate MTTs of a nested paramo catchment system in Ecuador with the purpose to tease out dominant controls on water transit time. The authors were able to identify relatively short transit times (< 1yr) compared to other environments in different climatic regions. The MTTs in their study site are mainly controlled by the catchment slope in relation to the dominant wetland soils. The experimentally derived dataset for this tropical ecosystem is unique and interesting to the HESS readership and beyond. The analysis is mostly sound and the paper generally well-written and structured.

Having said that, the paper struggles in parts to clearly convey the main points in line with the objectives of the study and could be shortened. I am missing a discussion around arguments that the MTT is not a meaningful catchment descriptor and the recent tendency towards the recognition of the time-variant nature of transit times. I do think that there are merits in using the MTT to characterize catchment systems particularly considering the constraints and limitations working in tropical environments; it should, however, be more clearly argued. Furthermore, there are some model decisions that should be more clearly explained, which also likely leads to additional analysis strengthening the paper and its line of arguments. Nevertheless, I think this is nothing that cannot be fixed with a careful revision to improve clarity and focus of the paper and I therefore support publication of this paper with some revisions.

Reply: We appreciate Christian Birkel´s (R2) revision and his constructive suggestions to improve the scientific quality of the manuscript and look forward to publishing this work in HESS, hoping to improve the understanding of hydrologic processes in tropical ecosystems. We agree that the paper could be shorter to focus its content in the objectives. We also acknowledge that some details of the modeling procedure deserve clarification along with a discussion about recent theoretical frameworks that explicitly incorporate time-variant transit times (please see response to R1). Our responses to R2 comments are outlined below.

We also want to highlight here, the major changes incorporated into the manuscript in relation to R2 suggestions. First, in the revised version of the manuscript we acknowledge the time-variant nature of MTTs (in P. 4, L 22-27) and justify why the LCA methodology (assuming steady-state conditions) is applicable for our study site in P. 5, L. 11-14 and P. 9, L. 16-19. Secondly, to shorten the paper and focus on its main objectives we removed two figures (Figures 3 and 6). Figure 3, the deuterium-oxygen18 relations for precipitation and stream waters, was removed as the information it conveyed was not relevant for the discussion of the paper and a thorough discussion of such information can be found in the manuscript of Mosquera et al. (2016), currently under review and accepted with minor revisions in Hydrological Processes. As recommended by R2 below, we also shortened the results and discussion regarding the probability and cumulative density functions in P. 16, L. 16-24 and removed Figure 6 as this do not add relevant information to the paper's discussion. Finally, we split section 3.2 (Model selection and mean transit time evaluation) into two shorter ones focused in different points. Section 3.2 (TTD evaluation and selection) regarding the selection of the best TTD and section 3.3 (Baseflow MTT) regards to the estimation of MTTs across the basin.

Specific comments:

Abstract:

Line 21: I'm not sure if the paper is about streamwater MTT as you excluded high-flow events from the analysis.

Reply: We agree, based on the discussion below, it is clear that MTT estimations correspond to baseflow MTTs. This is been specified in the abstract in P. 1, L. 24 and P. 2, L. 10 and throughout the rest of manuscript.

Key words:

Line 15: I suggest to simplify and reduce the key words to attract more online search results, e.g.: Ecohydrology, MTT, runoff generation, Andean páramo, Histosols, Ecuador.

Reply: We agree with the suggestion and have simplified and reduce the key words to the following: Ecohydrology, MTT, runoff generation, wet Andean páramo, tropical wetlands, Histosol, Ecuador

Introduction:

Page 3, Line 2: This is true for Latin America, but there are a few more studies in the tropics. You could even refer to Muñoz-Villers and McDonnell (2012) in this context.

Reply: We agree and have included additional references (Farrick and Branfireun, 2015; Muñoz-Villers et al., 2015) of studies recently conducted in the tropics in P. 3, L. 4-5.

Page 3, Line 17: I will come back to this point, but I think it's very likely that there's also a considerable near-surface runoff component as seen in other environments (you refer to Scotland and Sweden below) with organic rich wetland soils that remain saturated for much of the year. I, however, don't know the paper in review you cite here.

Reply: We see that this point can cause confusion. However, by "shallow subsurface flow" we refer to water moving in the first 30-40 cm within the soil matrix (i.e., "shallow organic horizon of the páramo soils located near the streams"). This is supported by water isotopic concentrations observed in the shallow organic horizon of the Histosol soils and stream waters in the study area. These observations and further discussion are presented in a manuscript which has been recently accepted with minor revision in Hydrological Processes receiving minor comments. We have provided the editor of this paper with a copy of the referred manuscript to pass it along to the reviewers.

Page 3, Line 29: This isn't entirely true, I'm afraid, because the dominating runoff generation process based on various tracer studies is a rapid near-surface flow. The subsurface component is a deeper and slower groundwater flux. Therefore, the wetland contribution can be quantified very well in form of near-surface saturation overland flow.

Reply: We partially agree with this comment. Although it is true that "rapid near-surface flow" has been observed in other environments, it mostly refers to "near-surface saturation overland flow". However, the hydrologic functioning of this particular system takes place in the first 30-40 cm of the soils, here referred to as "shallow subsurface flow", and saturation excess overland flow rarely occurs in the Zhurucay basin (i.e., high flows occur less than 3% of the time at the study site, see figure 3 in Mosquera et al., 2015). Thus, the term "rapid near-surface flow" does not apply to our system. We therefore have replaced the term "subsurface" by "shallow subsurface and/or near surface (i.e., overland flow)" to clarify this in P. 4, L. 5.

Page 4, Line 9: I think Broxton et al. (2009) worked in Arizona, USA. You could also specify the control you are referring to as in this case it was "aspect".

Reply: We appreciate the suggestion. We have corrected the site where this study was conducted and specified the controls on MTT in P. 4, L. 15-17.

Study site:

Page 5, Line 8: This is an awkward sentence, please revise.

Reply: We agree and have restructured the sentence in P. 6, L. 15-17.

Line 10: seasonality, primarily.

Reply: These suggestions have been incorporated in P. 6, L. 8.

Line 23: please, spell out INV.

Reply: INV is the name of the mining company (http://www.invmetals.com/about/history/).

Page 6, Line 4-5: Please, revise this sentence.

Reply: The sentences referring to soil distribution in the catchment have been restructured in P. 7, L. 12-15.

Line 27: Please, indicate model and make of the equipment.

Reply: Model and make of the instrument (Schlumberger DI500) are indicated in P. 8, L. 9.

Methods:

Page 9, Line 26: …is based…

Reply: We agree with this suggestion and have corrected it accordingly in P. 11, L. 9.

Line 27: I'm not sure I follow the second point.

Reply: This refers to input (recharge) function of the precipitation tracer composition to take into account recharge (i.e., volumetrically weighted isotopic composition) (McGuire and McDonnell, 2006). This has been clarified in P. 11, L. 10-11.

Page 10, Line 6: In this case, I suggest to consistently refer to a baseflow MTT and not streamwater MTT.

Reply: We agree. The following has been added in P. 11, L. 19-20: "As a result, estimations correspond to baseflow MTTs, hereafter simply referred to as MTT."

Page 11, Line 2-5: I fully agree that you seek to identify the best-performing and most parsimonious model. However, you don't really compare the models using a criterion for model selection (e.g., AIC, BIC or adjusted R2) that penalizes the number of parameters in combination with a goodness-of-fit measure. The MI criterion looks at how identifiable one parameter is, but not at the combined effect of more than one parameter used to calibrate the model.

Reply: We appreciate this comment. We applied the AIC metric for model selection as stated in P. 12, L. 11-14. Results from this analysis are reported in table 5 and P. 14, L. 21-24.

Page 11: How were models generated? Using a uniformly sampled Monte Carlo procedure?

Reply: This is correct. The fitting procedure included two steps for each model. 1) Initially 10,000 sets of parameter values were evaluated considering a wide range of parameter values sampled according to a uniform Monte Carlo procedure (Beven and Freer, 2001). The parameter ranges were wide. For instance the parameter range of the MTT of the EM model varied between 0 and 130 biweeks (5 yrs). 2) After the initial 10,000 runs, the range of the set of parameters that displayed relatively well identified were narrowed and the model was run again until 1,000 behavioral parameter sets were obtained (i.e., sets of parameters that yielded solutions corresponding to at least 95% of the highest KGE). This has been clarified in P. 12, L. 4-5.

Line 16: mainly?

Reply: We appreciate the suggestion "Majorly" is been changed by "mainly" in P. 13, L. 5.

Results:

Page 12, Line 12: Runoff coefficients show…

Reply: This sentence has been updated accordingly in P. 13, L. 27.

Page 13, Line 4-26: I'm not convinced by some of the statements present in this paragraph. For example, the best-fit gamma model compared to the best-fit exponential model does show a quite significant increase in performance (from 0.63 to 0.75) that can justify the use of one additional fitting parameter. On the other hand, a third fitting parameter resulted in an increased performance of only 0.01. The poorest model seems to be the DM with a best-fit of KGE=0.5. Based on this, one could qualitatively reject the DM and TPLR models as suitable models compared to the EM and GM. However, the decision between the EM and GM models should be informed by a model selection criterion such as the AIC (see comment above) that evaluates the combined effect of the parameters on model performance.

Reply: We completely agree and appreciate the suggestion. As indicated above we have now conducted an AIC evaluation and reports its results in Table 5 and P. 14, L. 21-24.

Page 14, Line 12: Please, revise this sentence.

Reply: Thank for the suggestion, sentence has been revised and corrected in P. 15, L. 11-12.

Page 14: I think that large parts here could be moved into the discussion or simply be deleted as later sections pick up on these issues. This would allow to shorten the m/s focusing on presenting the key results and later discussion in the light of the wider literature.

Reply: We agree with this suggestion. We have trim down irrelevant text in this section. See P. 15, L. 13-21.

Page 15, Line 26: just use MTT

Reply: This has been corrected accordingly in P. 16, L. 25.

Page 16, Line 5-12: Please, separate this very long sentence into smaller parts.

Reply: We appreciate the suggestion. This sentence has been split into three in P. 17, L. 3-10.

Discussion:

Page 17, Line 26: more depleted?

Reply: This was removed from the manuscript.

Page 18, Line 1: I think it would be better to indicate that baseflow MTT was analysed.

Reply: We agree. Changed in P. 18, L. 23.

Line 3: identifiability?

Reply: Correct. Changed in P. 18, L. 25.

Line 20: Was TMCF previously defined?

Reply: Thank you catching this. TMCF defined in P. 19, L. 13 and abbreviation consistently used thereafter.

Line 30: remain.

Reply: We agree. Corrected in P. 19, L. 24.

Page 19, Line 10: You somehow have to convince me that this actually is subsurface stormflow. I haven't seen the in review paper you mention in this context and all the evidence you show tells me that the dominating runoff generation mechanism is near-surface saturation overland flow due to little mixing with deeper soil horizons, short MTTs, etc.

Reply: Based on the characterization of the weekly isotopic composition of stream and soil waters conducted over a two-years period, it is evident that the isotopic composition of the shallow organic horizon of the Histosol soils consistently matches that of the streams, and that precipitation has essentially no direct influence in the streamflow isotopic composition (Mosquera et al., 2016). As such, even "subsurface stormflow" appears to inappropriately describe the system´s functioning, as water is preferentially delivered from the shallow 30-40 cm of the organic horizon of the Histosols to the streams regardless of the precipitation dynamics. Moreover, saturation excess overland flow rarely occurs at the study site (Mosquera et al., 2015) as stated in P. 3, L. 23-24. Therefore, we consider that "shallow subsurface flow" is indeed the appropriate term to define the delivery of water from the Histosols situated at the bottom of the slopes to the streams. As mentioned above, we have provided the editor with a copy of the cited paper currently in revision after minor changes have been addressed.

Line 20: You previously said up to 2.2 years in this context.

Reply: We referred to two different studies conducted in central Mexico. Muñoz-Villers and McDonnell (2012) reported MTTs of three years and recently Muñoz-Villers et al. (2015) reported MTTs ranging between 1.2-2.2 years. Therefore, this statement in the manuscript is correct.

Page 20, Line 6: I think Hrachowitz et al. (2009) argued with saturation overland flow.

Reply: Hrachowitz et al. (2009) actually reported that at the Lord Arch catchment runoff generation shows a flashy catchment response "dominated by runoff processes in the upper soil horizons." That is, in the 40 cm depth peaty soils, overlaying the mineral horizons,

Line 11: solutes.

Reply: Agree. Corrected in P. 21, L. 3.

Page 22, Line 1: explain.

Reply: Agree. Corrected in P. 23, L. 4.

Line 8: Please, revise this sentence.

Reply: We agree. Sentence is been corrected in line P. 23, L. 11-13.

Line 14: Isn't this simply the slope?

Reply: Yes, you are correct. Changed in lines P. 23, L. 17.

Line 24: I find the "regulation capacity" is coming a bit out of nowhere. What exactly do you mean by this? Is it in the sense of resilience or simply that the turn-over is quick and what goes in comes out with little delay?

Reply: We acknowledged how this can lead to confusion. Basically the páramo is an ecosystem recognized for its high discharge regulation capacity (i.e., páramo generates runoff year-round regardless of variability in precipitation inputs to the system). This characteristic is essential to the sustainability of human activities of downstream populations. However, little is known about the factors driving this regulation capacity. The results from this study provide information that improves our understanding of catchment functioning by identifying some of these drivers (see P. 1, L. 11-13 and P. 23, L. 23-28). That is, the interplay between soil storage and topography. We have explained the regulation capacity notion of the ecosystem in the introduction section of the manuscript in P. 3, L. 17-20.

Page 23, Line 2: It's the first time that you mention that SOF wasn't previously observed in the study catchment. This information needs to come earlier. I also think this whole paragraph can be shortened towards the key messages presented at the very end.

Reply: We agree. We have added information regarding SOF earlier in the manuscript in P. 3, L. 23-24. We have trimmed this paragraph to reduce its length in P. 23, L. 17 to P. 24, L. 2.

Line 29: Please, revise this sentence.

Reply: We agree, we have reworded the sentence in P. 24, L. 22-23.

Tables:

Shouldn't the current Table 2 come before you present the models (Table 1)?

Reply: We appreciate this suggestion and completely agree. Order of tables has been changed.

Current Table 1: I'm a bit confused about some decisions concerning the choice of initial parameter intervals. Why was the upper limit of tau set at 200 biweeks? This makes 2800days and over 7 yrs of TT, something stable isotopes aren't able to detect anyways (Stewart et al., 2010). Further, why was the lower limit of beta (GM) set to 0.5? In the case of low TT this could be well below 0.5 and on a global scale the average resulted to be at around 0.5 (Godsey et al., 2009). With the current lower limit in place you potentially miss suitable parameters that would also result in lower MTTs compared to current best-fit results; an argument you used to reject the GM. Also, it seems odd to me that you don't report the parameter interval for beta as this is the parameter you calibrate. The MTT (tau) is only the result of beta*alpha.

Reply: We really appreciate this comment. We used the MTT parameter ranges suggested by Timbe et al., (2014). However, we recognize that these authors had a different objective in their study and that it is reasonable that we constrain our parameter values range for MTT up to 5 years (130 weeks) (McGuire and McDonnell, 2006). As such, we have run all models again for all catchments, and statistics and figures have been updated accordingly. Regarding the parameter in the GM, we believe R2 refers to the alpha parameter. The alpha parameter lower limit was originally set up at 0.01, and the 0.5 value was just mistakenly reported in the table. We have corrected the lower limit value in the table and also reported the parameter range considered for beta.

Table 5: Similar issue here with the GM. I suggest to report the parameters alpha and beta.

Reply: Same as above. Beta parameter is now reported in the table.

Table 7: R2-values of 0.62 did not result significant? However, there's a relationship with flow characteristics particularly for the extremes and the runoff coefficient does seem to explain some of the spatial variability among catchments.

Reply: We appreciate this comment. The mentioned relations are not statistically significant at a 95% confidence level. Results are as follows:

Runoff coefficient:     $R^2$ 0.62          p-value: 0.39

Q99:          $R^2$ -0.42          p-value 0.24

Q10:          $R^2$ -0.61          p-value 0.12

Q5:          $R^2$ -0.62          p-value 0.11

P-values are now shown in table 7 and the relation between low flows and MTT is reported in P. 17, L. 18-21.

Figures:

Figure 2: What's the purpose of the streamflow inlet box? Could you not just show a log-scale to emphasize the low flow periods? Those event samples do show quite a bit of response to rainfall. What's the effect of pooling these out? Quite a bit shorter MTTs? Please, consider adjusting the different EC sampling period for comparison purposes.

Reply: 1) Streamflow inlet box: The purpose of the streamflow inlet box is to emphasize the response of low flows to rainfall inputs during the less humid periods. The box indicates flashy response even during these periods. We therefore still believe that the non-log-scale representation of the hydrograph in combination with the inlet box provides the best impression of the observed dynamics. 2) Event samples: The model runs reported were originally conducted once these referred event samples were pooled out from the streamflow isotopic composition time series. 3) EC sampling period: Sampling period for EC has been adjusted to hydrometric and isotopic data sampling period.

Figure 5: Please, clarify if sampling was started below alpha = 0.5 (GM) contrary to the information from Table 1. Again, I suggest to present the parameters alpha and beta.

Reply: We apologize for the confusion. The lower limit of the alpha parameter was originally set up at a value of 0.01. This was updated in Table 2. We now present both alpha and beta parameters as suggested, together with the MTT.

Figure 6: Is EPM missing in the right panel?

Reply: No, it is not. It just plots behind the EM curves in both panels. A note is been added to the caption of the figure.

Figure 8: If the MTT is normalized shouldn't it be unitless?

Reply: You are correct. We changed the text accordingly in figures 8 and 9.

References I used:

Godsey, S. E., Aas, W., Clair, T. A., de Wit, H. A., Fernandez, I. J., Kahl, J. S., Malcolm, I. A., Neal, C., Neal, M., Nelson, S. J., Norton, S. A., Palucis, M. C., Skjelkvåle, B. L., Soulsby, C., Tetzlaff, D. and Kirchner, J. W. (2010), Generality of fractal 1/f scaling in catchment tracer time series, and its implications for catchment travel time distributions. Hydrol. Process., 24: 1660 1671. doi: 10.1002/hyp.7677.

Stewart MK, Morgenstern U, McDonnell JJ. 2010. Truncation of stream residence time: How the use of stable isotopes has skewed our concept of streamwater age and origin. Hydrological Processes 24: 1646–1659.

REFERENCES WE USED:

[revised manuscript text omitted]

---

## Author Response (AR2)

**Point-by-point response to the Referee 1's review**

GENERAL COMMENTS

I appreciate the authors' kind and detailed replies and I think the paper is now improved. Still, there are in my opinion some misunderstandings on the concept of Mean Transit Time (MTT) that I will try to explain with more detail. I think the authors had a good point in limiting the analysis to baseflow conditions, but some more efforts could be put to clarify the meaning of their analyses.

Reply: We are glad to hear that the reviewer finds the paper is improved and appreciate his/her constructive suggestions to further improve our manuscript's scientific quality.

-The MTT is the mean of a distribution and as such it has some very useful mathematical properties. When the distributions are skewed, the mean is strongly affected by the "heaviness" of the tail, which is typically quite pronounced in catchments [see Kirchner, 2000]. The tail of the distribution is very difficult to estimate, especially in systems where the output signal is not very informative. This can be because the output is very damped with respect to the input, or because the input is characterized by regular fluctuations around a constant trend. Both of these are typical situations with water stable isotopes. In these cases, one may well be able to describe the measured solute response with a number of different functions which have the same 'young' component but very different 'old' components (or tails). Unfortunately, this has a huge impact on the mean of the distribution, which then remains very uncertain. Some authors have started to describe other properties of the transit time distributions (TTDs) other than the mean, like the median transit time [e.g. Soulsby et al., 2015], or the "young water fraction" [e.g. Kirchner et al., 2016ab].

Reply: As has been stated in the manuscript (P. 9, L. 17-21; P. 11, L. 9-20), we are aware of the mathematical properties of the transit time distributions (TTDs) used for modeling MTTs through the lumped convolution approach, as well as the advantages and limitations of this methodology. Because of those, the selection of the TTD distribution that best describes our system was conducted through a detailed statistical analysis to evaluate model's performance for each TTD in combination with available information about catchment functioning and runoff generation at the study site.

As the referee mentions, other authors have attempted to describe alternative properties of TTDs. However, Soulsby et al. (2015) do not provide theoretical basis for justifying the use and analysis of the median of TTDs instead of the MTT. The "young water fraction" metric proposed by Kirchner (2016a, 2016b) is on the other hand introduced with a detailed theoretical basis which should definitely be explored by catchment hydrologists. However, it is out of the scope of our current work to incorporate such analysis. We have clearly specified and justified the use of the LCA for estimating MTTs in our páramo catchment (P. 11, L. 9-20).

-Given the point above, it is clear that estimating catchment TTDs is not a trivial task, and uncertainty is often inevitable. For this reason, I am a bit skeptical about the fact that the exponential distribution is the most appropriate to describe the pàramo catchments. In my opinion, such a distribution is better identified because it has not enough degrees of freedom and the same distribution is used to fit short-term and long-term output features. As the short-term features are the only one providing information, they dominate the identification of the function. However, I would argue this is an "apparent" identification because it explains stream chemistry fairly well and parsimoniously, but is not able to reliably estimate the older water components, which have a large impact on the MTT. Other functions tested by the authors are less determined, in my opinion, because they allow a de-coupling between short-term and long-term information. Indeed, both the GM and TPLR predict MTT which are less constrained (296-1478 days and 356-1738) but also much larger than the EP (166-224 days). Doesn't this suggest that there may be a larger old water contribution than the one predicted by the EP? To further explore this problem it may be interesting to compare the results in terms of other metrics (e.g. are the median or the young water fraction similar in the EP and GM models?).

Reply: We agree that selecting catchment TTDs is not a trivial task. Given that, we did not only rely on the results of the detailed statistical analysis conducted for selecting the TTD that best describes the system under study, but also on the process-based knowledge of catchment functioning at the study site (Mosquera et al., 2015, 2016). These authors have determined that the Zhurucay catchment is a fast response system, where runoff generation is mainly controlled by water flowing in the shallowest 30-40 cm of the low developed páramo soils (~ 1 m depth) underlain by a young and compact geology. Additionally, these authors concluded that deep-groundwater contribution (old water) to discharge is minimal at the Zhurucay catchment. Our MTT estimates corroborate their findings. Although the contribution of old water component/s cannot be definitely neglected, it seems unfeasible that given the biophysical features of the landscape and its pedological and geomorphological structure, baseflow MTTs on the range of those yielded by the GM and TPLR TTDs are reliable. A note about this issue has been added in P. 15, L. 3-9. Therefore, we are confident that the results yielded by the EM provide robust baseflow MTT estimates in this páramo catchment. Regarding your last point, as we have already mentioned above, although comparing our results with those yielded by other metrics (e.g., the young water fraction) would definitely be interesting, that is out of the scope of this study.

-Finally, the idea that one single TTD exists is misleading. Any natural hydrologic domain responds differently before, during and after being forced by a storm event. This is not just about seasonality, it's also about each storm that crosses the catchment after the considered precipitation event and makes transit time distributions intrinsically non-smooth curves. Moreover, stream concentration is affected by the specific hydrologic conditions that take place at the time of sampling, and not by the average catchment conditions. This of course does not mean that stationary analyses are meaningless, but it should be clear that they try to depict an average catchment condition, which is often very different from any actual situation.

Reply: We agree that hydrologic systems are temporally dynamic and that such behavior needs to be accounted in models to better understand system's response. Although the LCA lacks to characterize such dynamics, assumptions of this methodology are quite clear and as such, we have carefully evaluated them in light of the conditions in our study site (). Here, we again want to clarify that our study aims to set a baseline for the application of modeling techniques using stable water isotopes in tropical ecosystems above the tree line, and that one of our future goals using this dataset includes the application of modeling techniques that explicitly recognize non-stationarity and storage dynamics in the hydrological behavior (e.g.,Birkel et al., 2015; Harman, 2015; Hrachowitz et al., 2013). However, those are out of the scope of our manuscript.

Overall, I don't want to look too critical about this work, but I think the paper would benefit from acknowledging a more realistic view on transit time distributions.

Reply: We agree and appreciate the referee's point of view. As so, based on the suggestions of both reviewers in the prior revision of our manuscript, we have already acknowledged the more realistic view on TTDs in light of the time-variant nature of MTTs (P. 4., L. 11-P. 5, L. 11). As such, we considered this point has been already addressed in our manuscript.

Other general comments are:

-as the AIC criterion was introduced, which takes into account the number of parameters, I think the Measure of Identification (MI) could be removed from the analysis, as it is redundant and applied rather arbitrarily given the lack of prior knowledge on the parameters.

Reply: We agree. The MI has been removed from the manuscript.

-I think it may be worth reducing the discussion section, which is currently very long and include the repetition of concepts which were already mentioned in the results.

Reply: We believe, that the readers will strongly benefit from the comprehensive discussion in its current form. Based on the suggestion from the reviewer, the section was again thoroughly reviewed and few small changes have been included to clarify the main ideas in the discussion though.

DETAILED COMMENTS

Abstract: it may be worth to specify from the beginning that the MTT refers to baseflow conditions

Reply: This was already specified in the abstract in P. 1, L. 24.

Page 4, line 7: please specify that you mean MTT variability among catchments

Reply: Thanks for the suggestion. Updated accordingly in P. 4, L. 7.

P. 4, l. 23: the first source of non-stationarity which should be mentioned is precipitation variability

Reply: We agree. Precipitation variability is now mentioned in L.4, P. 23.

P. 4, l. 27: what do you mean by stating that the cited works "have yielded results with high degree of uncertainty"? I personally think those works just show the complexity of the problem in more realistic terms.

Reply: We agree that those studies show the complexity of the problem. As such, results yielded from them provide high uncertainties as the methods used yet provide low capability to represent reality. This, to our view, is an important limitation of the used methodologies. Such uncertainty is highly dependent on the usually poorly recorded temporal variability of the input hydrometric and tracer field measurements, as we have already acknowledged in P. 5, L. 2-7. Therefore, we consider that the statement above is justified and appropriate in the context used in the manuscript.

P.5, l.3: what is a mathematical limitation?

Reply: We mean mathematical limitations to represent systems' non-stationarities. This has been updated in P. 5, L. 3.

P.9, l.19: I think this is a good point to mention that, as you do a steady-state analysis, you limit your study to the transit times of baseflow.

Reply: We agree. Updated accordingly in L. 9, P. 16.

P.11, l.15: I think this is a too strong sentence, which needs more context.

Reply: We agree. The sentence has been updated in P. 11, L. 15-16.

P.12, l.14: as mentioned in the general comments, I don't think the MI is necessary anymore.

Reply: We agree. The MI has been removed from the manuscript.

P. 12, l.25-27: first you say that total runoff is "spatially more heterogeneous", but then you say that Runoff coefficients show "relatively low spatial variability". This sounds inconsistent.

Reply: We think you mean P.13, L. 25-27. You are correct. Thanks for catching this inconsistency. We have updated the text accordingly in P. 13, L. 22-23.

P.14, l.16: I think the threshold on the KGE is critical for the parameter uncertainty. Could the author observe important differences with a different values? I am also wondering whether the models with a higher number of parameters would require a higher threshold.

Reply: We see your point about the critical threshold selection of the KGE value for parameter uncertainty. As such, we used a threshold already published in hydrologic science literature. We considered the threshold of 0.45 used by Timbe et al. (2014), who assed uncertainties in MTT estimations using the lumped convolution approach. Using different threshold values would be somehow arbitrary. Regarding your last point about models with higher number of parameters, we have already accounted for that in our model selection evaluation by using the AIC metric (P. 14, L. 16-20).

P.14, l.21 (Figure 3): I thinks there is no need to show the isotopic content of precipitation in this figure, as it was already presented in figure 2b. Moreover, such broad Y-axis makes it difficult to assess modeled and measured 18O in streamflow. I think the authors could reduce the Y-axis to e.g. [-8, -14]. Please also avoid the use of biweeks at the x-axis.

Reply: We agree. The figure has been replaced to show the observed and modeled isotopic composition in streamflow only. Biweeks has also been replaced by months in the x-axis.

P.15, l.9-10: "rather than from a more realistic representation of the hydrologic functioning of the catchments". This is a very strong and not justified statement.

Reply: We agree that this statement is not clearly justified. As so, we have decided to delete this sentence (see P. 15, L. 3).

P. 16, l. 20-24: I had already commented on this, but I see that we probably did not understand each others. First, the unit "biweeks" is rather unusual and I recommend using days (or months, or years) instead. As the exponential function integrates to 1 only in the limit to infinity, the notion of "completely recovered" is linked to a threshold (e.g. 99%) and I suggest to mention it. Finally, I don't fully agree with the last sentence. Even in a 3 years dataset, the first months are strongly affected by hydrologic events that had happened before the beginning of the measurements.

Reply: Thanks for the recommendation. We have changed the unit biweeks to weeks in the text. You are correct that the notion of completely recover might be ambiguous, and that mentioning a threshold is more appropriate. This has been updated in P. 16, L. 19-22. Lastly, we were careful to minimize the effect of prior hydrologic events before the beginning of the measurements. To minimize this effect, and has been done by other authors (e,g., Hrachowitz et al., 2011; Timbe et al., 2014), we looped the available three years of isotopic data ten times during calibration in order to extend the data series for 30 years as a warm-up period as stated in P. 11, L., 20-22.

ADDITIONAL LITERATURE
Kirchner, J., Feng, X., & Neal, C. (2000). Fractal stream chemistry and its implications for contaminant transport in catchments. Nature, 403(6769), 524–7. http://doi.org/10.1038/35000537

REFERENCES WE USED

Birkel, C., Soulsby, C. and Tetzlaff, D.: Conceptual modelling to assess how the interplay of hydrological connectivity, catchment storage and tracer dynamics controls nonstationary water age estimates, Hydrol. Process., 29(13), 2956–2969, doi:10.1002/hyp.10414, 2015.

Harman, C. J.: Time-variable transit time distributions and transport: Theory and application to storage-dependent transport of chloride in a watershed, Water Resour. Res., 51(1), 1–30, doi:10.1002/2014WR015707, 2015.

Hrachowitz, M., Soulsby, C., Tetzlaff, D. and Malcolm, I. A.: Sensitivity of mean transit time estimates to model conditioning and data availability, Hydrol. Process., 25(6), 980–990, doi:10.1002/hyp.7922, 2011.

Hrachowitz, M., Savenije, H., Bogaard, T. A., Tetzlaff, D. and Soulsby, C.: What can flux tracking teach us about water age distribution patterns and their temporal dynamics?, Hydrol. Earth Syst. Sci., 17(2), 533–564, doi:10.5194/hess-17-533-2013, 2013.

Kirchner, J. W.: Aggregation in environmental systems – Part 1: Seasonal tracer cycles quantify young water fractions, but not mean transit times, in spatially heterogeneous catchments, Hydrol. Earth Syst. Sci., 20(1), 279–297, doi:10.5194/hess-20-279-2016, 2016a.

Kirchner, J. W.: Aggregation in environmental systems – Part 2: Catchment mean transit times and young water fractions under hydrologic nonstationarity, Hydrol. Earth Syst. Sci., 20(1), 299–328, doi:10.5194/hess-20-299-2016, 2016b.

Mosquera, G. M., Lazo, P. X., Célleri, R., Wilcox, B. P. and Crespo, P.: Runoff from tropical alpine grasslands increases with areal extent of wetlands, CATENA, 125, 120–128, doi:10.1016/j.catena.2014.10.010, 2015.

Mosquera, G. M., Célleri, R., Lazo, P. X., Vaché, K. B., Perakis, S. S. and Crespo, P.:

Combined Use of Isotopic and Hydrometric Data to Conceptualize Ecohydrological Processes in a High-Elevation Tropical Ecosystem, Hydrol. Process., doi:10.1002/hyp.10927, 2016.

Soulsby, C., Birkel, C., Geris, J., Dick, J., Tunaley, C. and Tetzlaff, D.: Stream water age distributions controlled by storage dynamics and nonlinear hydrologic connectivity: Modeling with high-resolution isotope data, Water Resour. Res., 51(9), 7759–7776, doi:10.1002/2015WR017888, 2015.

Timbe, E., Windhorst, D., Crespo, P., Frede, H.-G., Feyen, J. and Breuer, L.: Understanding uncertainties when inferring mean transit times of water trough tracer-based lumped-parameter models in Andean tropical montane cloud forest catchments, Hydrol. Earth Syst. Sci., 18(4), 1503–1523, doi:10.5194/hess-18-1503-2014, 2014.

**Point-by-point response to Christian Birkel (Referee 2)'s review**

I had the opportunity to read the revised version of the manuscript "Insights on the water mean transit time in a high-elevation tropical ecosystem" by Mosquera et al. and found the paper much improved. The authors have extensively discussed and addressed all my comments, clarified my initial doubts about some aspects of the model approach, process interpretations and how parsimony was assessed, included new and significant analysis (re-run models, AIC calculated), and now they refer to the more up-to-date discussion around time-variable transit times (their incredibly fast responding system makes for an interesting contrast to most studies). The paper now makes for an excellent contribution to the literature about a geographical region with relatively sparse data and research. Therefore, I fully support publication of this paper.

All the best,

Christian Birkel

Reply: We are glad to hear that the reviewer finds that all his comments have been satisfactory addressed. We thank him for his constructive suggestions to improve the scientific quality of the manuscript and look forward to publishing this work in HESS hoping to improve the understanding of hydrologic processes in tropical ecosystems.

The only suggestion left is that the sentence on page 14, line 21 sounds rather odd and could be reformulated.

Reply: We agree with the suggestion. This sentence has been rewritten in P. 14, L. 16-17.

[revised manuscript text omitted]